



# Impact of information presentation on interpretability of spatial hazard information: Lessons from a study in avalanche safety

Kathryn C. Fisher[1], Pascal Haegeli [1], Patrick Mair[2]

[1]School of Resource and Environmental Management, Simon Fraser University, Burnaby, V5A 1S6, Canada
[2]Dept. Psychology, Harvard University, Cambridge, 02138, United States

*Correspondence to*: Pascal Haegeli (pascal_haegeli@sfu.ca)

**Abstract.** Avalanche warning services publish avalanche condition reports, often called avalanche bulletins, to help backcountry recreationists make informed risk management decisions about when and where to travel in avalanche terrain. To be successful, the information presented in bulletins must be properly understood and applied prior to entering avalanche
terrain. However, few avalanche bulletin elements have been empirically tested for their efficacy in communicating hazard information. The objective of this study is to explicitly test the effectiveness of three different graphics representing the aspect and elevation of avalanche problems on users' ability to apply the information.

To address this question, we conducted an online survey that presented participants with one of three graphic renderings of avalanche problem information and asked them to rank a series of route options in order of their exposure to the described
hazard. Following completion of route ranking tasks, users were presented with all three graphics and asked to rate how effective they thought the graphics were. Our analysis dataset included responses from 3,056 backcountry recreationists with a variety of backgrounds and avalanche safety training levels. Using a series of generalized linear mixed effects models, our analysis shows that a graphic format that combines the aspect and elevation information for each avalanche problem is the most effective graphic for helping users understand the avalanche hazard conditions because it resulted in higher success in
picking the correct exposure ranking, faster completion times, and was rated by users to be the most effective. These results are consistent with existing research on the impact of graphics on cognitive load and can be applied by avalanche warning services to improve the communication of avalanche hazard to readers of their avalanche bulletins.

## 1 Introduction

Snow avalanches are a serious threat that destroys property and claims the lives of people in mountainous regions around the
world every year. While catastrophic avalanches hitting mountain villages are responsible for the largest number of fatalities in mountain ranges such as the Himalayas, most avalanche deaths in western countries involve individuals heading into avalanche terrain for recreation. In North America, for example, avalanches claimed the lives of 334 recreationists between 2011 and 2020 (Avalanche Canada, 2019; CAIC, 2020), and even though there are no reliable statistics, it is suspected that many more recreationists are caught in avalanches but manage to escape the most severe outcome. While a small number of



affected individuals were guides or ski patrollers professionally engaged in managing the avalanche risk for paying guest or clients, the vast majority were lay people making their own decisions about when and where to recreate in the backcountry.

When travelling in the backcountry avalanche risk is managed by carefully assessing the nature and severity of the hazard using weather, snowpack and avalanche observations (e.g., McClung, 2002). This assessment must be combined with additional information about the terrain exposure of an intended backcountry trip to the avalanche hazard to make an

informed decision about whether going ahead with a trip is acceptable to the individual under the observed conditions. Under most circumstances, recreationists are responsible for completing this complex assessment without professional guidance to inform their decisions. To assist recreationists with understanding the existing avalanche hazard conditions and making these assessments, most western countries have established avalanche warning services that publish daily condition reports— commonly known as avalanche bulletins, forecasts, warnings, or advisories—that summarize the current snowpack and

avalanche situation across predefined forecast areas. These reports are intended to give recreationists the information needed to make an informed risk assessment of a planned backcountry trip.

While the specific design of avalanche bulletins differs from country to country, they all present the information in a tiered structure that is referred to as the "information pyramid" (EAWS, 2021). At the top of the pyramid is the avalanche danger rating, which describes the overall severity of the avalanche conditions using the signal words and colors of the ordinal, 5-

level avalanche danger scale. The 5-level scale was introduced in 1993, and while there are subtle differences between the European and North American versions (EAWS, 2018; Statham et al., 2010), it is the cornerstone of public avalanche risk communication around the world. The next level of the information pyramid describes the nature of the avalanche hazard in more detail. Over the last decade, the concept of avalanche problems has established itself as a useful framework for explaining the nature of avalanche hazard in a structured way. Avalanche problems represent actual avalanche risk

management concerns that can be described in terms of their type, location, likelihood and size of avalanches. In North America, the conceptual model of avalanche hazard (Statham et al. 2018a) defines nine different avalanche problem types, and avalanche bulletins describe the nature of up to three active avalanche problems using a combination of iconic graphics and text. European avalanche warning services utilize a smaller list of avalanche problem types, and even though conceptually similar, use less formalized terminology to explain the location and nature of the present problems. The next

level of the information pyramid provides users with more detailed but still synthesized overviews of existing weather conditions, relevant snowpack structures and avalanche activity observations. Some avalanche warning services also include links to raw data such as weather, snow profile or avalanche observations in their bulletins. These observations are the foundation of the hazard assessment presented in the bulletin and represent the final level of the information pyramid. The intent of the pyramid is to present information about a complex hazard in an easily accessible and concise way while

allowing users with greater information needs and more advanced skills to explore more details.

Avalanche warning services belong to a wider range of warning services and government agencies whose mandate is to communicate information about a complex and spatially variable natural hazard to the public in a meaningful way. Weather forecasters and local governments routinely issue statements to communities faced with fire, flood or storm watches and





warnings. In these disciplines, considerable attention has been paid to improving risk communication products by testing
which elements of risk communication messages are effective and which may lead to unintended consequences (see, e.g.,
Cuite et al., 2017; Morss et al., 2016; Rickard et al. 2017). For example, research into storm surge messaging identified that
recipients that saw messages about extreme storm surges were more likely to express intentions to evacuate, but also were
more likely to rate the information as more overblown and the source less reliable (Morss et al., 2016). Similar efforts to
empirically test the effectiveness of warning messages and safety signage are underway in the outdoor recreation field (e.g.,
Saunders et al., 2019; Weiler et al., 2015) to provide managers with evidence-based guidance on how to communicate with
their visitors.

Recognizing the crucial importance of the avalanche bulletin for the safety of backcountry recreationists, the avalanche
safety community has recently started to examine its effectiveness more systematically. These efforts can be divided into
three main research themes. Several recent projects have examined the quality and consistency of the information presented
in avalanche bulletins as providing accurate hazard information is crucial for effective risk communication (Lundgren and
McMakin, 2018). Example studies of this research theme include Lazar et al. (2016) who presented public avalanche
forecasters with a series of avalanche danger scenarios to see whether they interpret them the same, Techel et al. (2018) who
examines the spatial consistency and bias of avalanche danger ratings in avalanche bulletins in the European Alps, Statham
et al. (2018b), who studied the consistency of avalanche problem assessments among the warning services in the Canadian
Rocky Mountains, and Clark (2019) who studied the link between avalanche problem assessments and danger ratings in
Canadian avalanche bulletins. All of these studies highlighted considerable challenges and the need to improve the
production of avalanche bulletins.

The second and equally important research theme is trying to better understand how backcountry recreationists use and apply
the information provided in the avalanche bulletin. The risk communication research community has stressed for a long time
that having a good understanding of the target audience is a critical prerequisite for effective risk communication (Lundgren
and McMakin, 2018). Traditionally, the avalanche safety community has classified avalanche bulletin users simply
according to their preferred activity (e.g., backcountry skiing, mountain snowmobiling, snowshoeing), level of formal
avalanche awareness training (none, introductory course, advance level course, or professional level training), and/or basic
sociodemographics. Winkler and Techel (2014), for example, used data from two online surveys to determine who uses the
Swiss avalanche bulletin and how these users have changed over time. More recently, St. Clair (2019) conducted a
qualitative interview study to better understand how winter backcountry recreationists use, understand and apply the
avalanche bulletin information in their avalanche risk management process. Her analysis revealed a sequence of five distinct
bulletin information use patterns that incorporate increasingly more complex information and are able to manage avalanche
risk at higher levels of sophistication. This typology provides a valuable framework for evaluating the effectiveness of risk
messages with respect to the types of decisions that the users are intending to make. St. Clair's study was followed up by
Finn (2020) who conducted a large-scale online survey to examine whether bulletin users who say they use the avalanche
bulletin at a certain level also have the necessary skills to do so effectively. Finn's results offer valuable insight into



avalanche bulletin literacy at the different levels of St. Clair's bulletin user typology and highlights user groups that might have misconceptions about their skill levels.

The third theme of avalanche bulletin research is the explicit examination of its effectiveness. Empirically testing how messages resonate with users and whether they result in the desired behavioural response is an important but challenging part of risk communication research. Example of these types of studies in the avalanche field include Burkeljca (2013a, 2013b), who examined the usability of four different avalanche bulletin products (Canada, Catalonia, Tyrol and Utah) using a small sample of 14 that included lay people and experts from Slovenia. Winkler and Techel (2014) examined the results from the

same two surveys mentioned previously to shed light on how the complete revision of the Swiss avalanche bulletin in 2014 affected users' perceived quality and usability of the product. Similarly, Engeset et al (2018) conducted an online survey to better understand the effectiveness of the Norwegian avalanche bulletin. This study explicitly asked participants about their preferences for different forms of information presentation (text, symbols, or pictures) and empirically assessed users' comprehension of two hazard situations as a function of the type and amount of information presented. The authors used

both the appropriateness of the risk management approaches chosen by participants and their self-reported effectiveness rating to assess the efficacy of the avalanche hazard descriptions.

Since assessing the suitability of backcountry trips requires recreationists to relate the information provided in the bulletin to the terrain characteristics of their intended trips, the description of the spatial distribution of avalanche hazard within a forecast area is a crucial component of the avalanche bulletins. While there is considerable complexity in how avalanche

hazard interacts with terrain (e.g., Bühler et al., 2013; Bühler et al., 2018), the primary location information included in avalanche bulletins focuses on elevation and aspect. However, current avalanche bulletin products exhibit substantial variability in what the elevation and aspect information refers to and how it is presented. Swiss avalanche bulletins, for example, state a single danger rating for a forecast region and the accompanying aspect and elevation information highlights the core zones where the stated avalanche danger applies the most (SLF, 2020). The French avalanche bulletins use the same

approach as the Swiss (MeteoFrance, 2021), whereas the Norwegian bulletins also just publish a single danger rating per forecast region, but aspect and elevation information is used to describe where the identified avalanche problems are most prevalent (Varsom, 2021). The recently launched Euregio avalanche bulletin publishes elevation specific avalanche danger ratings and also provides aspect and elevation information for each of the existing avalanche problems (EAWS, 2021). Most avalanche bulletins in North America publish avalanche danger ratings for different elevations and describe the location of

avalanche problems with respect to elevation and aspect. While the elevation descriptions in European avalanche bulletins are generally specific (e.g., above 2200 m) and change daily depending on conditions, North American bulletins use predefined elevation bands (alpine, treeline or near treeline, below treeline) to specify avalanche danger and the location of the avalanche problems.

In addition to these differences in the use of elevation and aspect information, there are also different styles on how this

information is presented. While most of the European and Canadian avalanche warning services use separate graphics for communicating aspect and elevation information, the warning services in the United States and New Zealand use so-called



aspect-elevation rose diagrams that show the elevation and aspect information together in a single graphic (NZAA, 2021; USFS, 2021). Within each of these groups, we can find slight variations in design. The aspect-elevation rose diagrams of the Northwest Avalanche Center and the Colorado Avalanche Information Center are straight octagons with grey shading, the aspect-elevation rose of the New Zealand avalanche warning service has an extra corner in each aspect segment and the shading reflect the danger rating of the elevation band, and the Utah Avalanche Center used a three-dimensional aspect-elevation rose diagram (CAIC, 2021; UAC, 2021; NWAC, 2021; NZAA, 2021).

The goal of this study is to contribute to our understanding of the efficacy of avalanche bulletins by empirically testing the effectiveness of individual components. Our starting point is the fact that a multitude of graphics are used by avalanche warning services around the world to communicate avalanche problem characteristics. Several studies have demonstrated that graphics used might not be well understood and users struggle to combine the information when making terrain choices (e.g., Burkeljca, 2013a; Burkeljca, 2013b; Engeset et al., 2018; Finn, 2020). To better advise avalanche warning services on which graphics are most effective with users, we conducted an online survey to experimentally test if altering the presentation format of the location information of avalanche problems can improve users' ability to apply it to hypothetical terrain choices. The results of this study help warning services to improve their avalanche bulletin design so that recreationists can make better informed choices about when and where to travel in the backcountry.

## 2 Methods

In the spring of 2020, we conducted a large-scale online survey to empirically examine different options for improving the presentation of location information in North American avalanche bulletins. The three main questions that the survey aimed to shed light on were:

a) How does the presentation format of the avalanche problem location information (i.e., aspect and elevation) affect users' ability to apply this information when assessing the exposure of routes to avalanche hazard?

b) Can adding an interactive exercise help improve users' ability to apply the avalanche problem location information?

c) How well do the travel advice statements included in avalanche problem section of North American avalanche bulletins resonate with users?

The focus of this paper is to present the insight we have gained about the first research question. The results that relate to the other two questions are described in separate manuscripts.

### 2.1 Survey Design

To systematically test whether the presentation format of the avalanche problem location information affects users' ability to apply the information, our survey included a series of route ranking task where participants were presented with an avalanche bulletin with two avalanche problems and a custom-built topographic map with three routes (Figure 1). The terrain map depicted a simplified mountainscape with slopes of consistent incline on all aspects and elevation bands. The task of





participants was to study the avalanche bulletin information and then rank the three depicted routes according to their exposure to the described avalanche problems. The correct solution for the ranking task could be determined by counting the

number of aspect and elevation segments each route crossed where avalanche problems were present. The more avalanche problem aspect and elevation segments a route crossed, the more exposed it was to avalanche hazard. Participants were explicitly alerted that overhead hazard and terrain traps should not be included in their assessment.

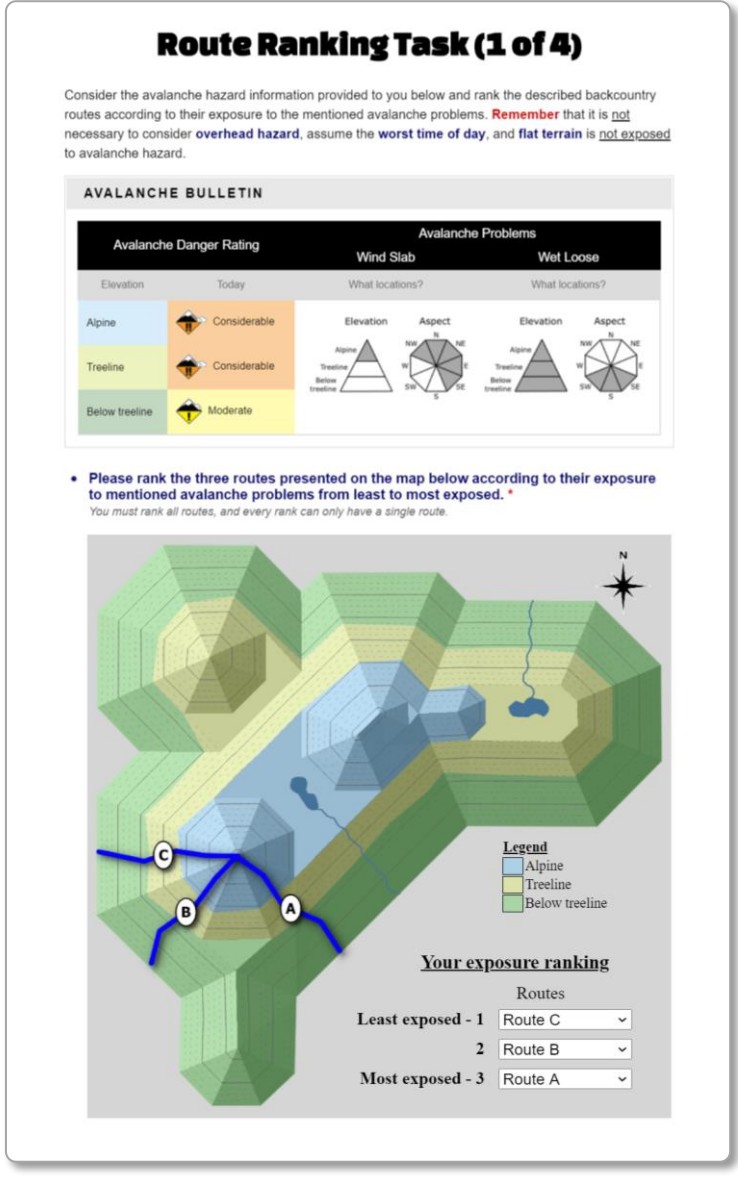

**Figure 1: Example of route-ranking exercise with avalanche bulletin scenario and custom-built topographic map with three simple routes.**





In our experiment, the avalanche problem information was presented in one of three graphic formats (Figure 2). The first format had aspect and elevation information separated for each avalanche problem similar to the graphic used in Canadian avalanche bulletins, while the second format had aspect and elevation combined into a single aspect-elevation rose graphic for each avalanche problem like in the US bulletins, and the third format presented the aspect and elevation information for all avalanche problems combined. Throughout the rest of this paper, we will refer to these three presentation formats as *Separate*, *Aspect-Elevation Rose*, and *Combined*. To prevent the specifics of the avalanche bulletin information to affect our results in unintended ways, our experiment included six different avalanche bulletin scenarios (see Appendix), all of which were developed in conjunction with avalanche industry experts to ensure they represent realistic real-world conditions.

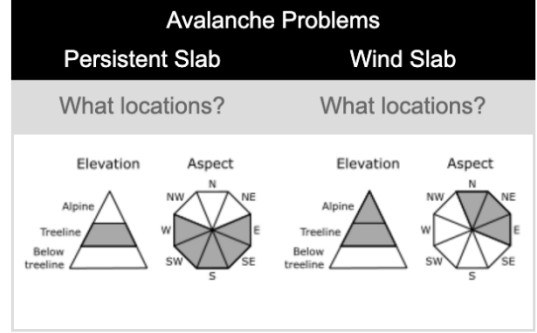

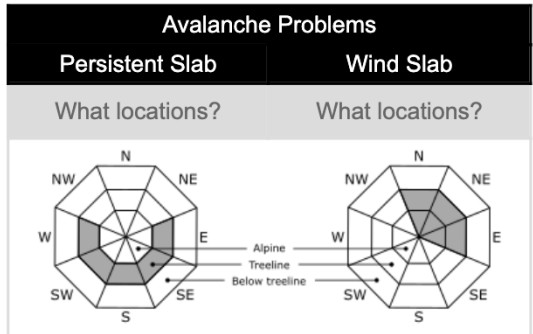

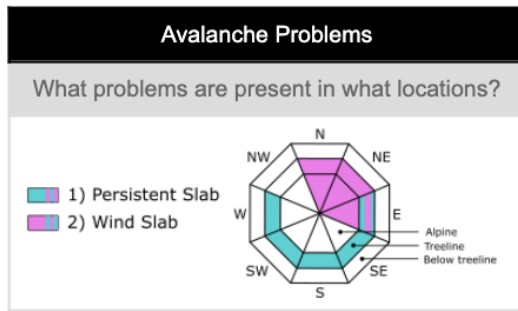

**Figure 2: Presentation formats for location information of avalanche problems: Separate graphics (top panel), Aspect-Elevation Rose diagram (middle panel), and Combined graphic (bottom panel)**



Each survey participant was presented with two random avalanche bulletin scenarios using one of the three aspect and elevation information presentations, and they completed two route-ranking exercises for each of the bulletin scenarios. The
first ranking exercise for each bulletin scenario included "simple" routes that crossed only one aspect, whereas the second exercise had "complex" routes that crossed multiple aspects. Between the two avalanche bulletin scenarios, participants were presented with a range of different feedbacks to examine how an interactive exercise can affect participants' ability to apply the avalanche problem information to terrain. However, this part of the experiment is not the focus of this manuscript. In summary, the experimental portion of the survey included four route-ranking tasks that were complete in the following
sequence:

1) Avalanche bulletin scenario 1 – Simple routes
2) Avalanche bulletin scenario 1 – Complex routes
3) Feedback (none, articulate process, solution, solution with explanation)
4) Avalanche bulletin scenario 2 – Simple routes
5) Avalanche bulletin scenario 2 – Complex routes

After completion of the route-ranking tasks, participants were shown all three avalanche problem information graphics and asked to rate their effectiveness for communicating the location information of avalanche problems on a scale from 0 (not effective at all) to 100 (extremely effective). In addition, participants were given the opportunity to provide additional comments in a text box.

To contextualize the results of the route-ranking exercise and the effectiveness ratings, our survey included a wide range of background questions. We drew from questions included in Finn's (2020) survey and asked participants to indicate their primary modes of recreating in the backcountry, which avalanche bulletin region they recreate in, how often they check the bulletin, how many years and days per year of experience they had, what their overall attitude towards avalanches is, the level of avalanche training they had completed, and their bulletin user type as described by St. Clair (2019). Additional
questions asked participants to identify how much weight they ascribe to different bulletin sections and rate their confidence in their abilities to understand the bulletin, recognize hazardous conditions in the field, make safe choices, and read topographic maps. Also included in the survey was a question explicitly testing users topographic map reading skills, as well as basic sociodemographic questions including self-identified gender, age, education level, location of residence, and colorblindness.

The survey was developed during the early part of the 2019/20 winter season and extensively tested in February and March 2020 prior to release. Survey testing began with an initial round of testers with moderate to high levels of winter backcountry recreation experience and avalanche industry experts. A second round of testing included users from novice to expert participants. The survey was also reviewed and approved by the Office for Research Ethics of Simon Fraser University (SFU ethics approval 2020s0074).





## 2.2 Recruitment and Survey Development

The primary target audience for our survey was North American avalanche bulletin users, which we recruited in a variety of ways. The foundation of our recruitment were 3047 bulletin users who participated in previous avalanche bulletin surveys conducted by our research program and indicated that they were interested in participating in future studies. The survey was officially launched on March 23, 2020 by sending invitation emails to 300 individuals from this existing panel of prospective participants. This soft launch allowed us to monitor the initial responses and address any survey issues if necessary. However, the survey worked as designed and no modifications were required. On March 26, 2020, we sent invitation emails to the rest of our panel of prospective participants (2747 individuals) and between March 26 and April 1, 2020 the survey was also actively promoted by our partnering avalanche warning services (Avalanche Canada, Parks Canada, Colorado Avalanche Information Centre, Northwest Avalanche Center). Each of these warning services helped us recruit participants by including a banner on their bulletin website and promoting the survey through their social media channels. We also advertised our study by posting on various social media sites popular among winter backcountry users, such as *South Coast Touring* and *Backcountry YYC* on Facebook, and by reaching out to community leaders to distribute the survey among their followers.

To ensure meaningful and even samples for each of the experimental treatments included in our survey (type of location information graphic, type of feedback), participants were stratified according to their preferred winter backcountry activity and bulletin user type before being assigned to one of the experimental treatments. This guaranteed that all treatment combinations had representation from each winter backcountry activity and bulletin user type even if they were relatively small.

The survey sample for the present analysis was drawn on May 31, 2020, after which no additional surveys were included in analysis. At the close of the survey 6789, individuals had started our survey and 3668 (55.3%) completed it. The vast majority of the dropouts (1829, 58.6% of dropouts) did not continue after looking at the first page of the survey that described the objective of the study and structure of the survey. The dropout rate for individual survey pages was 1% or less except the page that introduced the route-ranking task (57, 3.4%). Of the individuals who completed the survey, 1600 (44.6%) were participants of previous survey studies of our research group who received an invitation email. Other substantial recruitment sources included announcements on avalanche bulletin websites (17.5% of participants who completed survey), social media posts by collaborating avalanche warning services (9.2%), and other posts in social media groups (e.g., Facebook, Instagram) focused on winter backcountry recreation (21.5%).

## 2.3 Data Analysis

To assess the effectiveness of the three different aspect-elevation graphics in a meaningful way, we focus on a triad of performance measures:

- The correctness of participants' answers in the route-ranking exercise,



- Participants' completion time of the route-ranking exercise, and
- Participants' perceived effectiveness of the three graphics

with an initial hypothesis that a more effective presentation would be associated with a higher percentage of correct answers,
quicker completion times and higher perceived effectiveness ratings.

This combination of measures provides a comprehensive perspective on the effectiveness of the different graphics that builds on existing research into the role of cognitive load in the success of different graphic types. Response time and response accuracy of primary and secondary tasks was used by Dindar et al. (2015) to measure the cognitive load of static and animated graphics on students learning English. The authors additionally used self-reported cognitive load as an additional

metric to estimate cognitive load. In this study, we replaced the subjective, explicit request to estimate cognitive load with a question asking about perceived effectiveness. We also focused our study on a single type of task because of our interest in directly measuring how the graphic influences application of bulletin information. Our single-task approach is similar to Martin-Michiellot and Mendelsohn (2000) who measured response time and assessment accuracy in relation to different computer manual presentation formats.

Our analysis approach started with the use of standard descriptive statistics to describe the nature of the analysis dataset and explore the relationships between different variables. The core of our analysis consisted of three generalized linear mixed effects models (GLMM) that explored the three different performance measures outlined above. GLMMs are an extension of generalized linear models that properly account for the correlations that emerges from repeated measure designs or nested data structures (Harrison et al., 2018; Zuur et al., 2009). To accommodate these data structure, GLMMs include both fixed

and random effects in the regression equations. The fixed effects, which are equivalent to the intercept and slope estimates in traditional regression models, capture the relationship between the predictor and response variables for the entire dataset. While traditional regression models assign the remaining unexplained variance in the data (i.e., randomness) entirely to the overall error term, mixed-effect models partition the unexplained variance that originates from groupings within the dataset into random effects. Thus, random effects highlight how groups within the dataset deviate from the overall pattern described

by the fixed effects included in the model. While there is some judgment involved in deciding what predictors are included in a GLMM as a fixed or random effect, it is generally the grouping variables that are not explicitly of interest that enter the analysis as random effects.

To assess how the graphics influence participants' ability to complete the route-ranking task correctly, their responses were graded as follows. Participants who ordered all three routes correctly received a passing grade whereas all other responses

were assigned a failing grade. This means that we ended up with a binary response variable, which we examined with a logistic mixed effects regression model that uses a logit link to model the relationship between a binary response variable and one or more predictors. The random effects included in this model were participant ID and the ranking task avalanche scenario.





To examine the effect of the graphics on completion time in seconds, we used a gamma mixed effects regression model,
which is suitable for a continuous, positive, potentially right-skewed response variable. Similar to the model for correctness, we included participant ID and ranking task scenario as random effects.

The third and last GLMM included in this analysis explored the relationship between the graphics and participants' ratings of perceived effectiveness. Since these ratings were on a bound scale from 0 to 100, we used a beta mixed effect regression model for this analysis (Cribari-Neto and Zeileis, 2010). Similar to the logistic regression model, a beta regression uses a
logit link to relate the response variable to the predictors in a constrained way. Prior to analysis, we divided participants' ratings by 100 to scale them down to 0 to 1 and transformed them with $y_{trans} = (y_{orig}(n-1)+0.5)/n$ as suggested by Smithson and Verkuilen (2006) to eliminate values that are exactly 0 or 1 since they cannot be handled by the beta regression. In this model, participant ID was the only random effect as each participant rated all three graphics but there were no scenarios.

Since assessing the impact of the graphic and how this effect might vary among different levels of avalanche training is the main objective of this study, the initial versions of all three models included the type of aspect-elevation graphic and participants' level of formal avalanche training as predictor variables (both as main and interaction effects). The correctness and completion time models also included the following variable describing the nature of the ranking task: complexity of the route options (simple or complex), whether it was the first or second set of route-ranking tasks, and what type of feedback
was provided between the two sets. In addition to these default predictors, the effects of other participant characteristics (e.g., primary winter backcountry activity, whether survey was completed on a smartphone, score on the map reading test) and route-ranking task attributes (e.g., overall number of correctly completed ranking tasks, which graphic was used in ranking tasks) were explored during the model building process. The predictors were only kept in the models if they contributed to the model as determined by a Type II Wald chi-squared test with a p-value smaller than 0.050 and the size of
their effects were meaningful. Differences between model variants were assessed with likelihood ratio tests, and BIC (Schwarz, 1978) and model interpretability were used to guide final model selection.

We conducted our entire analysis in R (Version 4.0.5; R Core Team, 2021) and used the glmmTMB package (Brooks et al., 2017) to estimate our mixed effects models. The Type II Wald chi-squared tests were calculated using the Anova function of the car package (Fox and Weisberg, 2019). To assess violations in model assumptions, we simulated quantile residuals
(Dunn and Smyth, 1996) as implemented in the DHARMa package (Hartig, 2020). Visual inspection of the resulting diagnostic plots (e.g., Q-Q-plot for uniformly distributed residuals) did not suggest any substantial model violations. Due to the logit link function and the presence of both main and interaction effects, the parameter estimates emerging from the regression models in this study are difficult to interpret directly. To make the results more tangible, we calculated marginal means of the response variables (i.e., correctness, completion time, perceived effectiveness) for the levels of different
predictor variables and followed up with post-hoc pairwise comparisons to assess whether these estimates were significantly different from each other. We performed this part of the analysis using the functions included in the emmeans package (Lenth, 2019). To counteract the issue of Type I error inflation from multiple comparisons, we calculated Holm-corrected p-





values. The results of these analyses are presented in so-called effects plots, which display the differences between levels of a predictor variable of interest while holding all other predictor variables constant at their base levels. Hence, it is more important to look at the differences between the attribute levels of the predictor variable of interest than the absolute values.

## 3 Results

### 3.1 Participant Demographics

To ensure meaningful results, we only included participants in our analysis dataset who completed all pages of the survey, whose reported residence was in Canada or the United States, who were over the age of 20, and whose choices for primary activity and avalanche awareness training aligned with the predefined options. In addition, we excluded participants who took less than 10 minutes or more than 2 hours to complete the survey, or who spent longer than 10 minutes completing the route ranking tasks or reading feedback between the tasks. These cut-offs were chosen after a visual inspection of the distribution of page viewing times and are expected to represent participants who either did not engage with the survey or got interrupted. The final analysis dataset consisted 3,056 participants, which represented 83.3% of the 3668 individuals who completed the survey. The median completion time of the survey was 24.6 minutes with an interquartile range of 18.5 to 32.6 minutes.

Of the 3,056 participants, 76.9% self-identified as male (2,328 participants), 36.9% (1,125 participants) were between 25 and 34 years old, and 79.8% had a university-or-higher education (2,426 participants). In terms of avalanche safety training, 46.9% (1,433 participants) had taken an introductory level recreational avalanche safety course, 18.9% (577 participants) an advanced level recreational course, and 16.4% (501 participants) had completed a professional training course. Backcountry skiers represented the highest proportion of recreationists in the study with 78.3% of the sample (2,394 participants) identifying backcountry skiing as their primary backcountry winter activity. The largest group of participants (31.3%, 955 participants) were relatively new to their sport, with 2 to 5 years of backcountry experience. However, the second largest group of participants (24.5%, 750 participants) had over 20 years of experience. Bulletin user types 'D—Distinguish Problem Conditions' and 'E—Extends Analysis' made up 75.6% of participants (2,312). Finally, 69.8% (2,134) of responses were from residents of the USA.

### 3.2 Correctness of participants' answers

Overall, our analysis dataset included 12,224 individual route-ranking tasks, of which 74.6% were completed correctly. Our final model for the probability of completing the route-ranking task correctly included seven fixed effects. The main effect for type of feedback as well as the interaction effects between graphic type and participants' level of formal avalanche training and the interaction effects between type of feedback and participants' level of formal avalanche training were eliminated due to p-values larger than 0.05 (Type II Wald chi-square test). The parameter estimates from the regression analysis are presented in Table 1, but the effects plots (Figure 3) show the key results in a more tangible way.


The avalanche problem information graphic that a participant saw during the task exercises had a significant main effect on
whether a participant completed the tasks correctly (Figure 3). Comparing the three information formats shows that
participants who saw the *Aspect-Elevation Rose* graphic were the most likely to complete the tasks correctly (probability =
0.752).[1] Participants who saw the *Combined* graphic had significantly lower probability (0.711, p-value < 0.008)[2] of
completing the tasks correctly than those who saw the *Aspect-Elevation Rose*. Similarly, participants seeing the *Separate*
graphic were less likely to complete the tasks correctly than those seeing the *Aspect-Elevation Rose* (0.722), but the
difference was statistically not significant (p-value = 0.085). Furthermore, there was no statistically significant difference in
the performance between participants who were presented with the *Separate* and *Combined* graphic (p-value = 0.775).

The level of avalanche training a participant had completed was also a significant predictor of completing the task correctly
(Figure 3). Participants with professional training had the highest probability of completing the task correctly (0.768)
followed by participants with advanced and introductory recreational-level training (0.739 and 0.737). The probability of
participants with no training completing the tasks correctly was 0.664. Our examination of the differences between
consecutive levels revealed that the difference between participants with no training and introductory level recreational
training was significant (odds ratio: 1.42; p-value < 0.001). The increase between recreational and professional level training
was not statistically significant (p-value = 0.259).

Additional factors that changed the probability of completing the tasks correctly included route type and task set. Participants
were more likely to complete tasks correctly with the simple routes than the complex ones (0.800 versus 0.643, p-value <
0.001), as well as during the second set of tasks rather than the first set (0.745 and 0.712, p-value < 0.001). Participants'
probability of completing the tasks correctly was also related to characteristics such as their primary backcountry activity,
success on the map reading task, and phone use. Within our sample, individuals who identified snowmobiling as their
primary activity were significantly less likely to complete the tasks correctly than backcountry skiers (0.656 versus 0.784, p-
value < 0.001). Snowmobile accessed backcountry skiers exhibited a similar pattern to snowmobilers, with a probability of
0.636 of completing the tasks correctly. Participants who passed the map test were more likely to complete the tasks
correctly than those who failed it (0.771 versus 0.682, p-value < 0.001). Participants who completed the survey on a phone
were less likely to complete the tasks successfully than those who used a desktop (0.711 versus 0.745, p-value = 0.005).

---

[1] All response variable values presented in the model section are calculated for the particular level of the predictor variable
of interest while holding all other predictor variables constant at their base levels.

[2] All p-values presented in the model sections are from post-hoc pairwise comparisons. They are Holm-corrected p-values to
counteract the issue of Type I error inflation from multiple comparisons.



**Table 1: Parameter estimates of regression model examining the correctness of participants' responses in the route-ranking exercise. Dashes (-) indicate that the level represents the base level of the attribute. (Number of observation = 12,224)**

| | | Parameter estimate | Standard error | p-value | p-value of Type II Wald Statistic |
|---|---|---|---|---|---|
| **Main effects** | | | | | |
| *Predictor* | *Level* | | | | |
| Graphic type | Separate | - | - | - | 0.0082 |
| | Aspect-Elevation Rose | 0.1564 | 0.0736 | 0.0334 | |
| | Combined | -0.0500 | 0.0734 | 0.4961 | |
| Avalanche training | None | - | - | - | <0.0001 |
| | Introductory | 0.3475 | 0.0774 | 0.0002 | |
| | Advanced | 0.3571 | 0.0942 | <0.0001 | |
| | Professional | 0.5152 | 0.0992 | <0.0001 | |
| Route type | Simple | - | - | - | <0.0001 |
| | Complex | -0.8008 | 0.0479 | <0.0001 | |
| Set number | First set of two | - | - | - | 0.0003 |
| | Second set of two | 0.1693 | 0.0468 | 0.0003 | |
| Map literacy | Fail | - | - | - | <0.0001 |
| | Pass | 0.4488 | 0.0606 | <0.0001 | |
| Primary activity | Snowshoeing | - | - | - | <0.0001 |
| | Ice climbing | 0.0432 | 0.2343 | 0.8537 | |
| | Out-of-bounds skiing | 0.1743 | 0.1541 | 0.2579 | |
| | Backcountry skiing | 0.2200 | 0.1230 | 0.0737 | |
| | Snowmobile-accessed backcountry skiing | -0.5146 | 0.2309 | 0.0258 | |
| | Snowmobiling | -0.4262 | 0.1648 | 0.0097 | |
| Response via phone | No | - | - | - | 0.0047 |
| | Yes | -0.1731 | 0.0613 | 0.0047 | |
| Intercept | | 0.9078 | 0.3013 | 0.0026 | |
| **Random effects** | | **Number** | **Variance** | **Std. Dev** | |
| Individual participant | | 3056 | 0.6818 | 0.8257 | |
| Avalanche problem scenario | | 6 | 0.4253 | 0.6521 | |


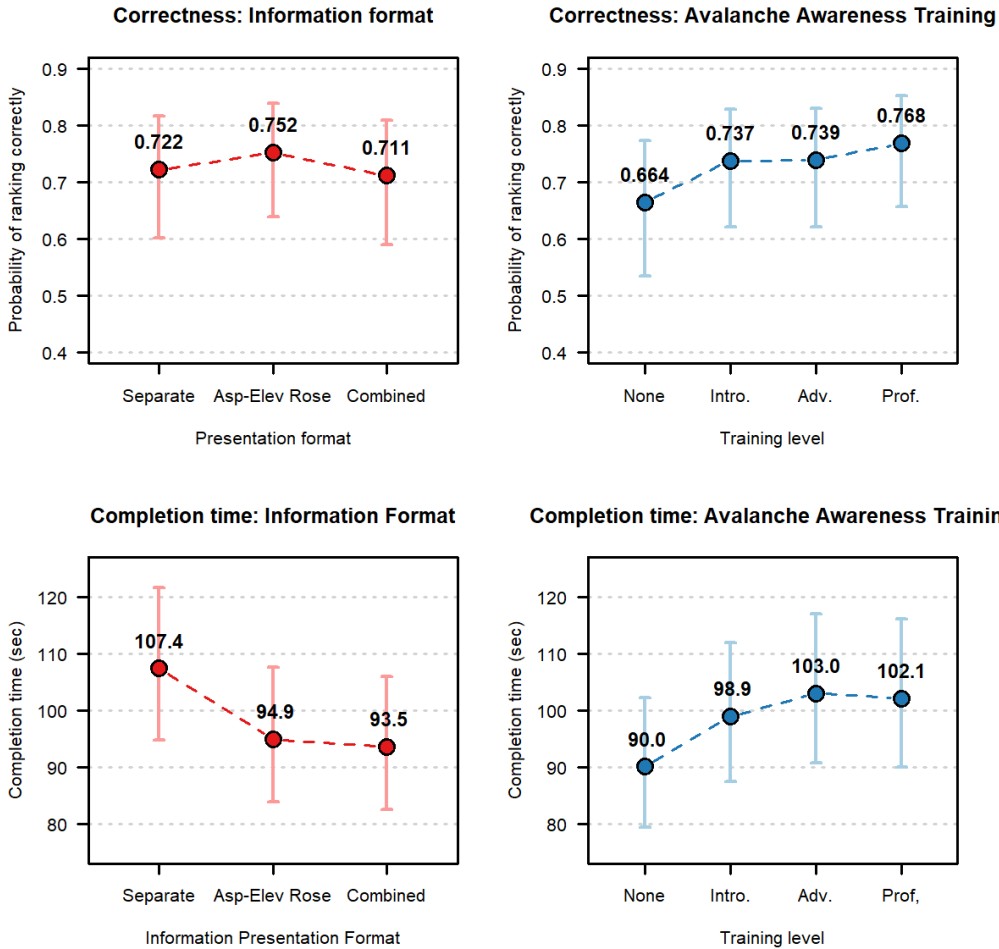

**Figure 3: Effects plots illustrating the main effect for the presentation format and avalanche awareness training levels in the correctness and completion time model. Error bars represent 95% confidence intervals for probability of ranking correctly and completion time calculated from the subsample for the particular parameter level.**

### 3.3 Participants' completion time

Participants took a median of 87.0 seconds to complete the route-ranking task exercises and the interquartile range of completion times was from 60.0-134.0 s. Our final model describing completion time of the task exercises included seven main effects, and individual participants and bulletin scenarios were included as random effects (Table 2). As in the correctness model, the interactions effects between graphic type and participants' level of formal avalanche training as well as between type of feedback and participants' level of formal avalanche training were eliminated due to p-values larger than 0.05 (Type II Wald chi-square test).





**Table 2: Parameter estimates of regression model examining the participants' completion time of the route-ranking exercise.** 385 **Dashes (-) indicate that the level represents the base level of the attribute. (Number of observations = 12,196)**

| | | Parameter Estimate | Standard error | p-value | p-value of Type II Wald Statistic |
|---|---|---|---|---|---|
| **Main effects** | | | | | |
| *Predictor* | *Level* | | | | |
| Graphic type | Separate | - | - | - | < 0.0001 |
| | Aspect-Elevation Rose | -0.1234 | 0.0202 | < 0.0001 | |
| | Combined | -0.1384 | 0.0203 | < 0.0001 | |
| Type of feedback | None | - | - | - | 0.0012 |
| | Articulate process | 0.0642 | 0.0207 | 0.0020 | |
| | Answers | -0.0137 | 0.0205 | 0.5035 | |
| | Answers & Explanation | 0.0164 | 0.0206 | 0.4276 | |
| Avalanche training | None | - | - | - | < 0.0001 |
| | Introductory | 0.0942 | 0.0217 | < 0.0001 | |
| | Advanced | 0.1347 | 0.0258 | < 0.0001 | |
| | Professional | 0.1260 | 0.0268 | < 0.0001 | |
| Route type | Simple | - | - | - | < 0.0001 |
| | Complex | 0.1178 | 0.0083 | < 0.0001 | |
| Set number | First set of two | - | - | - | < 0.0001 |
| | Second set of two | -0.1861 | 0.0150 | < 0.0001 | |
| Map literacy | Fail | - | - | - | < 0.0001 |
| | Pass | 0.1030 | 0.0172 | < 0.0001 | |
| Age category | Linear trend | 0.0900 | 0.0063 | < 0.0001 | < 0.0001 |
| Intercept | | 4.2820 | 0.0695 | < 0.0001 | |
| **Random effects** | | **Number** | **Variance** | **Std. Dev** | |
| Individual participant | | 3049 | 0.1337 | 0.3656 | |
| Avalanche problem scenario | | 6 | 0.0229 | 0.1512 | |

Our analysis revealed that the format of the avalanche problem information graphic had a significant effect on the completion time for route-ranking task (Figure 3). Based on the estimated model, participants who saw the information with aspect and elevation separate for each avalanche problem (*Separate*) took the longest time to complete the tasks (estimated 390 marginal mean 107.4 s). Participants who saw the *Aspect-Elevation Rose* or *Combined* graphic took significantly less time to





complete the tasks. The estimated marginal means for the completion time were 94.9 s (difference: -12.5 s; p-value < 0.001) for the *Aspect-Elevation Rose* and 93.5 s (difference: -13.9 s; p-value < 0.001) for the *Combined* graphics. The difference between the *Aspect-Elevation Rose* and *Combined* graphics did not emerge as significant (1.4 s; p-value = 0.0.725).

Our analysis also revealed a significant effect of the type of feedback participants received between the two sets of route
ranking exercises. Relative to receiving no feedback, participants who had to articulate their process, took significantly longer to complete the task (difference: +6.4 s; p-value 0.006), whereas receiving the solutions with or without explanations did not result in a significant difference in completion times (p-values: 0.817 and 0.752).

Avalanche training had a significant effect on completion time. In general, the more recreational level training participants had completed, the longer they took to complete the task. Based on the model, participants with advanced level recreational
training took the longest to complete the route ranking task (103.0 s; 13.0 s longer than participants with no formal training; p-value < 0.001), closely followed by participants with professional training who completed the tasks in 102.1 s (12.1 s longer than participants with no formal training; p-value < 0.001). Participants with introductory recreational levels training took 98.9 s (difference 8.9 s; p-value < 0.001), and participants with any training 90.0 s. This means that the biggest jump between consecutive categories occurs between no and introductory recreational-level training and effect diminishes with
higher levels of training.

Other factors that emerged as significant predictors of completion time include the experimental variables route type and the task set, as well as the participants' characteristics map reading test result and age. Participants ranking a scenario with complex routes took 11.6 s longer (p-value < 0.001) than when ranking simple routes. Conversely, participants were quicker at ranking the second set of routes than the first set (89.7 versus 108.0 s; p-value < 0.001). Participants who failed the map
reading test also complete the tasks substantially more quickly than participants who passed (93.5 versus 103.6 s; p-value < 0.001). Completion times increased linearly with the age category of participants with each increasing age class taking approximately 3 s longer (p < 0.001).

## 3.4 Perceived effectiveness rating

Our final regression model for the perceived effectiveness ratings included six main effects and three 2-way interaction
effects (Table 3). Across all participants, the highest ratings were given to the *Aspect-Elevation Rose* graphic, with an estimated marginal mean rating of 78.4 out of 100. This is significantly higher than either the *Separate* (71.7, p-value < 0.001) or *Combined* graphics (71.9, p-value < 0.001). There was no significant difference between the ratings for these two graphics (p-value = 0.973).

In addition to the overall effect of the information presentation format, there was also an interaction effect with a
participant's country of residence (Figure 4). Canadian residents gave nearly identical ratings for the *Separate* graphics (75.0) and the *Aspect-Elevation Rose* diagram (74.8), with no significant difference between them (p-value = 0.990). Canadian residents rated the *Combined* graphic the lowest of the three formats (71.7), which was not significantly different from the other presentation formats (p-value = 0.012 and 0.017, respectively). In contrast, US residents rated the *Aspect-*



**Table 3: Parameter estimates of regression model examining the participants' perceived effectiveness ratings. Dashes (-) indicate that the level represents the base level of the attribute. (Number of observations = 8,876)**

| Fixed Effects | | Parameter Estimate | Standard error | p-value | p-value of Type II Wald Statistic |
|---|---|---|---|---|---|
| **Main Effects** | | | | | |
| *Predictor* | *Level* | | | | |
| Graphic Type | Separate | - | - | - | <0.0001 |
| | Aspect-Elevation Rose | -0.5689 | 0.1205 | <0.0001 | |
| | Combined | -0.4881 | 0.1234 | <0.0001 | |
| Country of residence | Canada | - | - | - | 0.2989 |
| | USA | -0.3305 | 0.0500 | <0.0001 | |
| Avalanche Training | None | - | - | - | 0.2696 |
| | Introductory | -0.0990 | 0.0652 | 0.1130 | |
| | Advanced | -0.0717 | 0.0749 | 0.3382 | |
| | Professional | -0.0963 | 0.0783 | 0.2192 | |
| Used in task exercises | No | - | - | - | <0.0001 |
| | Yes | 0.5924 | 0.0479 | <0.0001 | |
| Tasks answered incorrectly | Linear trend | -0.0774 | 0.0220 | 0.0004 | 0.0169 |
| Completed on phone | No | - | - | - | 0.0002 |
| | Yes | 0.1157 | 0.0308 | 0.0002 | |
| Intercept | | 1.0410 | 0.0906 | <0.0001 | |
| **Interaction Effects** | | | | | |
| *Predictor (levels)* | *Predictor (levels)* | | | | |
| Graphic Type[a] | Country of residence | | | | <0.0001 |
| Aspect-Elevation Rose | Canada | - | - | - | |
| | USA | 0.7328 | 0.0672 | <0.0001 | |
| Combined | Canada | - | - | - | |
| | USA | 0.3478 | 0.0682 | <0.0001 | |
| Graphic Type | Avalanche Training | | | | 0.0068 |
| Aspect-Elevation Rose | None | - | - | - | |
| | Introductory | 0.1547 | 0.0835 | 0.0638 | |
| | Advanced | 0.1461 | 0.0998 | 0.1433 | |
| | Professional | 0.1977 | 0.1047 | 0.0590 | |
| Combined | None | - | - | - | |
| | Introductory | 0.0031 | 0.0851 | 0.9704 | |
| | Advanced | -0.0768 | 0.1020 | 0.1433 | |





| | | | | | |
|---|---|---|---|---|---|
| Graphic Type | Professional<br>Used in task exercises | -0.2145 | 0.1071 | 0.0452 | <0.0001 |
| Aspect-Elevation<br>Rose | No | - | - | - | |
| | Yes | -0.3103 | 0.0676 | <0.0001 | |
| Combined | No | - | - | - | |
| | Yes | 0.0171 | 0.0683 | 0.8025 | |
| Graphic Type | Tasks answered incorrectly | | | | <0.0001 |
| Aspect-Elevation<br>Rose | Linear trend | 0.1982 | 0.0294 | <0.0001 | |
| Combined | Linear trend | 0.1290 | 0.0300 | <0.0001 | |
| **Random Effects** | | **Number** | **Variance** | **Std. Dev** | |
| Individual Participant | | 3056 | 0.132 | 0.3633 | |
| Overdispersion parameter for beta family: 1.57 | | | | | |

[a] Base level is Graphic type = Separate

*Elevation Rose* diagram significantly higher (81.6) than either the *Separate* (68.3, p-value < 0.001) or *Combined* (72.1, p-value < 0.001) graphics. Unlike, Canadian residents, US residents rated the *Separate* graphic significantly lower than the Combined presentation format (p-value = 0.001).

In addition to the interaction effect above, there was also an interaction effect between the format of the avalanche problem graphics and a participant's completed level of avalanche awareness training (Figure 4). The ratings of the *Aspect-Elevation Rose* tended to increase with increasing levels of training. For participants who completed professional level training, the *Aspect-Elevation Rose* was rated 79.2 versus the *Separate* graphic at 71.1 (significantly different, p-value < 0.001) and for the *Combined* graphic it was 68.3 (significantly different from *Aspect-Elevation Rose* at p-value < 0.001, not significantly

different than *Separate* style p-value = 0.18). The difference in rating between the *Aspect-Elevation Rose* and other styles decreases at lower levels of training, showing that at lower levels of training the effect of the *Aspect-Elevation Rose* graphic is not as preferred over other formats. Among participants with no training, the difference between the *Aspect-Elevation Rose* and the *Separate* graphic was the smallest (77.4 versus 73.1; p-value 0.005), and no other differences were significant among this group.

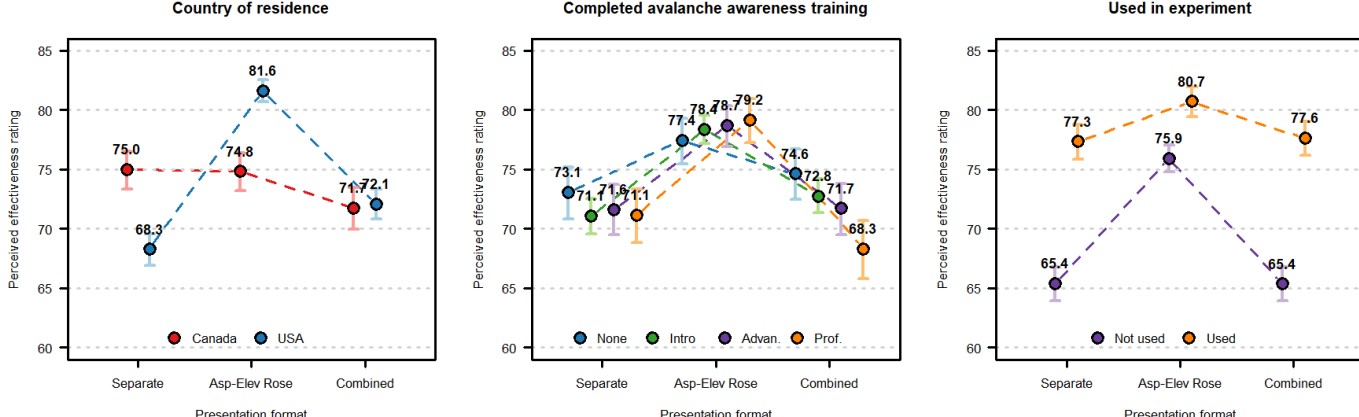

**Figure 4: Effects plots illustrating the interaction effects with presentation format in the perceived effectiveness rating model. Error bars represent 95% confidence intervals for perceived effectiveness calculated from the subsample for the particular parameter level.**

Another interaction effect was observed between the information presentation and whether a participant used it during the task exercises. Participant rated graphics they used during the task section of the survey higher than graphics they did not use during the survey (Figure 4). However, the difference in the rating for the graphics between participants who had not and who had used them was lower for the *Aspect-Elevation Rose* than for the *Separate* or *Combined* graphics. This shows that the *Aspect-Elevation Rose* graphic was rated higher than the other two graphics even when participants had no familiarity with the icon from previous use in the survey.

There was also an interaction effect between the format of the graphics and how well a participant performed during the task exercises. For the *Aspect-Elevation Rose* and *Combined* graphic, participants' ratings of the graphics tended to increase with the number of tasks they completed correctly. In contrast, ratings of the *Separate* graphic tended to decrease with the number of tasks a participant completed correctly.

Unlike the other models, only one additional explanatory factor contributed to explaining the variation ratings. Participants who used their phone overall rated all of the graphics just slightly more favourably (75.3 versus 73.0, p-value < 0.001).

## 4 Discussion

We defined the success of an avalanche problem location information graphic based on whether participants completed the ranking task exercises correctly, how long it took them to complete the task, and how highly they rated the perceived effectiveness of the graphics. The use of regression analysis allowed us to isolate the influence of the graphics on each of these three metrics by controlling for the other influencing factors.

We can present an overall picture of user experience with each graphic by looking at a combination of the three metrics described above. The *Separate* graphic led to lower rates of correct task completion, slower task completion times, and was





given relatively low ratings by all levels of training. Canadian residents rated the *Separate* graphic as about equivalently useful to the *Aspect-Elevation Rose* diagram, but US residents rated it the lowest of all the graphics. The *Separate* graphic

received low ratings when compared to the *Aspect-Elevation Rose* regardless of whether it was used in the task exercises or not. These results indicate that the *Separate* graphic has challenges communicating avalanche problem information and we suspect that its popularity among Canadian residents is likely due to familiarity.

The *Aspect-Elevation Rose* graphic led to the highest rate of correct task completion, fast completion times, and was given the highest rating by all levels of training. It received the highest ratings regardless of whether or not survey participants

used it during the task exercises, was rated by far the highest graphic by US residents and was considered equivalent to the *Separate* graphic by Canadian residents. These results indicate the *Aspect-Elevation Rose* diagram is an effective graphic for communicating avalanche problem information and is likely to be accepted by many users.

The *Combined* graphic led to lower rates of correct task completion, on par with the *Separate* graphic, but fast completion times. The *Combined* graphic received relatively low ratings by both Canadian and US residents, regardless of whether or

not it was used in the task. It received low ratings across all training levels, with ratings decreasing as training increased. These results indicated that the *Combined* graphic is not effective for communicating avalanche problem information, and not likely to be accepted by users.

### 4.1 Cognitive load perspective on results

Our results are consistent with existing research on the effect of cognitive load on task performance. According to cognitive

load theory, individuals have limited memory resources to apply to processing information, and that cognitive load increases with an increase in working memory use. Higher levels of cognitive load often lead to poor learning outcomes, lower task success, or trouble applying information (Allen et al., 2014; Dindar et al., 2015; Martin-Michiellot and Mendelsohn, 2000). Sweller et al. (2011) describe how cognitive load is altered by "interactivity", which refers to the elements that must be processed simultaneously to be understood. Higher levels of interactivity generally lead to higher cognitive load. The authors

further highlight that more information can be processed simultaneously when the information is broken down into meaningful "chunks" known as schema. Cognitive load can also be described as either intrinsic or extrinsic. Intrinsic cognitive load refers to the challenge inherent in understanding information or completing a task, whereas extrinsic cognitive load emerges from how the material is presented (Sweller et al., 2011). These two types of cognitive load are additive, with both competing for working memory capacity. If a task has a high intrinsic cognitive load, it is advised to reduce the

extrinsic cognitive load as much as possible, as studies have found that people struggle with making behavioral choices when information is presented in a cognitively demanding format (Allen et al., 2014). There are multiple strategies for estimating cognitive load that include performance on tasks, efficiency of task completion, and self-reported ratings of cognitive load—often in combination although the relationship between measurements varies under different conditions (Dindar et al., 2015; Sweller et al., 2011). In the avalanche safety context of this study, interpreting the avalanche problem

graphics and making the route choice selection both demand cognitive resources from participants. Based on this, we can





think of the metrics used to evaluate the problem graphics in this study as reflective of the cognitive load experienced during the task exercises. Completion of the route-ranking exercise is in itself an intrinsically challenging activity but did not vary between treatments, so it is expected that differences in outcome reflect the extrinsic cognitive load of the graphics.

The concept of extrinsic cognitive load helps explain the poor success of the *Separate* and *Combined* presentation formats. The *Separate* graphic is distinguished by a low success rate on the route ranking exercise, slow completion time, and low ratings for the graphic's perceived effectiveness. All of these indicators together suggest that the route-ranking exercise with this presentation format for the avalanche problem location information produced a high cognitive load that lead to poor performance. In this presentation format, users had to combine the aspect and elevation information for multiple avalanche problems. Each individual component of the graphic could only be applied to terrain once combined with the others, which means that had graphic had high element interactivity. We hypothesize that this high element interactivity led participants to focus their cognitive resources on interpreting the graphic and lowering the resources available for actually applying the information to the terrain and ranking the routes. Additionally, to integrate the information, users had to direct their attention to multiple locations in the graphic to make sense of the information. There is evidence that this kind of attention splitting also leads to a higher cognitive load on individuals (Martin-Michiellot and Mendelsohn, 2000; Sweller et al., 2011).

With evidence that integrated information should lead to reduced cognitive load, it would be expected that the *Combined* graphic would lead to the least cognitive load because it integrated the most information into a single graphic. However, our results show that users also had a high amount of difficulty applying the information from this presentation format to the route-ranking exercise as demonstrated by the low correctness scores despite faster completion times. This result may be due to the high visual complexity of the *Combined* graphic leading to a high extrinsic cognitive load for the graphic. The *Combined* graphic uses multiple colours to represent avalanche problems, and the meaning of the colours must be distinguished and interpreted to understand the information presented in the graphic. Complex visuals have been shown to be difficult to interpret as they increase users' extrinsic cognitive load (Anderson et al., 2011; Harold et al., 2020; Masri et al., 2008). Therefore, we suggest that that the extrinsic load from the complex visuals was high enough to reduce performance on the route-ranking exercise. Our results also mirror the result of studies on website complexity and hospital signage showing that visuals with medium levels of complexity performed most successfully with users (Rousek et al., 2011; Wang et al., 2014).

From a cognitive load perspective, the finding that the *Aspect-Elevation Rose* diagram performs best is not surprising. This presentation format mitigates the cognitive load required to integrate the avalanche problem aspect and elevation information by combining those elements into a single graphic—lowering element interactivity. However, it keeps the avalanche problems separate. This degree of integrating information may correspond well to users existing schema or mental model about avalanche danger. In North America, the conceptual model of avalanche hazard uses "avalanche problems" as a framework to organize information about avalanche hazard. In the conceptual model, "location" is identified as one of four main characteristics of avalanche problems and, at the bulletin scale, "location" is described by aspect and elevation (Statham et al., 2018). The success of the *Aspect-Elevation-Rose* graphic may be in part because it taps into this existing



conceptual framework for thinking about "location" as a single characteristic defining avalanche problems. The *Aspect-Elevation-Rose* graphic is the only graphic that represents "location" for each avalanche problem, and therefore most closely represents aspect and elevation as they are included in the conceptual model. In contrast, the *Combined* graphic—with it is combination of avalanche problems into a single graphic—aggregates location information at a higher level than is used in the conceptual model of avalanche hazard.

## 4.2 Implications for avalanche warning services

The results of this study offer valuable insights for avalanche warning services seeking to communicate avalanche problem information to users more effectively. Our findings indicate that the *Aspect-Elevation Rose* diagram leads to the best performance in the route-ranking task, indicating that this presentation format may be best suited towards helping recreationists use the information as part of the avalanche bulletin. The *Aspect-Elevation Rose* was the most effective across all groups, and even users who are accustomed to the Canadian-style graphic can benefit from the US-style graphic.

Our results show that avalanche warning services interested in changing their information presentation might initially find resistance from their users as users prefer graphics that they are already familiar with. The interaction between country of residence and preference rating for the graphics suggests that users hold favourable perceptions of whichever graphic they are most familiar with. However, users may be flexible and willing to accept new graphics after experience with the graphics. Comparing the preferences of users on a per-graphic basis, participants who saw the *Combined* graphic during the task exercises exhibited the greatest increase in rating compared to those who did not use it. This boost to the preference of the *Combined* graphics by participants who used it in the tasks suggests that it may take relatively little time for users to become accustomed to a change in avalanche problem information graphics. This suggests any resistance to changing graphics used in the bulletin may be short lived.

Other results from this study that may be of interest to avalanche warning services is the finding that avalanche education was a strong predictor of how successfully people completed the ranking task. We found that participants with recreational level avalanche awareness training performed similarly to those with professional level training regardless of which graphics they used—indicating that recreational training is successfully helping users interpret avalanche bulletins. This is consistent with prior research demonstrating that avalanche education is a significant factor influencing avalanche bulletin literacy (Finn, 2020). More importantly in the context of the objective of this study, however, our results show that the *Aspect-Elevation Rose* is the best presentation format for all training levels. Hence, there is no need to design different sets of graphics for beginners.

Additionally, this study found that participants with different primary backcountry activities performed differently on the task exercises even after controlling for avalanche awareness training. However, there was no interaction effect between the type of avalanche problem graphic used and participants' primary backcountry activity, indicating that the graphic use was not a factor in this variation of performance. Avalanche warning services can use this as evidence that changing avalanche problem graphics will not disadvantage backcountry recreationists of any sport. However, the route-ranking exercise may





have been optimized for backcountry skiers based on the route design, and further research is needed to determine if the effect of backcountry activity on the results could be eliminated by optimizing the route-ranking exercise for different

activities. Additional research is also needed to determine if the effects observed for during this desktop exercise can be translated into increased recognition of hazardous aspect and elevation combinations in the field.

Finally, the success of combining avalanche problem aspect and elevation into the *Aspect-Elevation-Rose* graphic opens new doors for further improvements to the avalanche bulletin. In addition to aspect and elevation, likelihood and size are two additional avalanche problem characteristics that are presented graphically in North American avalanche bulletins. While

likelihood and size are assessed and presented in a single chart in the conceptual model of avalanche hazard (Statham et al., 2018), the two characteristics are presented in separate graphics in North American bulletins. Since this study has demonstrated that there are benefits to linking conceptually related avalanche hazard information into a single graphic for public use in avalanche bulletins, future research should seek to identify if this principle could also be extended to present likelihood and size in a single graphic or if it would disadvantage users with low graphical literacy.

**4.3 Limitations**

The participant sample in this study demonstrates trends consistent with previous surveys of backcountry recreation users. A high proportion of university educated, male, backcountry skiers, between 25 and 34 years of age with basic avalanche education engage in online surveys about avalanche safety (Finn, 2020; Haegeli and Strong-Cvetich, 2020; Haegeli et al. 2012). The similarity in sample demographics may be drawn from the similar survey promotion techniques used between

this study and Finn (2020). Although this study and Finn (2020) did reach a wider range of users than previous studies, it only captures the behaviour of the demographic that responds to an online survey and may underrepresent non-English speaking participants or other demographics. Additionally, though the survey was open to snowmobilers, the task exercises were not optimized to show routes that would be realistic from the perspective of a recreational snowmobiler.

**5 Conclusion**

To make informed decisions about when and where to travel in the backcountry, winter backcountry recreationists need to manage their risk from avalanches by monitoring the hazard conditions and relating this information to the terrain characteristics of their intended trips. The daily avalanche bulletins published by local avalanche warning services provide critical information about the existing conditions when recreationists are planning their trips from home. We used an online survey to evaluate the impact of avalanche bulletin information graphics on participants ability to apply the information to a

route-ranking exercise that simulated the planning process for a backcountry trip. We evaluated the graphics on the correctness and completion times of user responses during the exercise, as well as useability ratings provided by users. Our study identified that combining aspect and elevation information into a single graphic leads to improved success on the



route-ranking exercise, quicker completion times, and is favored by users regardless of avalanche training experience or country of origin. These results can be used by avalanche warning services seeking to maximize useability of their bulletins.

This study highlights that simply changing the graphic presentation of the aspect and elevation information can lead to greater success in applying the information to a route-finding task. These research results also provide valuable insight for the presentation of hazard information beyond avalanches by demonstrating that linking graphical hazard information to existing mental models about the hazard can lead to better application of the information. This lesson may help to improve communication of any natural hazard warning information where applying graphic information is necessary to make safe

decisions.

*Code and data availability.* The data, code, and output for our analysis and the data and code for the figures and tables included in this paper are available at https://doi.org/10.17605/OSF.IO/MYFP2 (Haegeli et al., 2021).

*Author Contribution.* KF and PH designed and executed the study, and prepared the original draft. PM assisted with data analysis and reviewed the draft. All authors reviewed the draft prior to submission.

*Competing Interests.* PH is part of the editorial team of NHESS. The others authors declare that they have no conflict of interest.


*Acknowledgements.* The authors thank Avalanche Canada, the Colorado Avalanche Information Center, and the Northwest Avalanche Center for their input during the design of the survey and their promotion of the study among their communities. We are grateful to the Coast Salish peoples including the Tsleil-Waututh (səl̓ilw̓ətaʔɬ), Kwikwetlem (kʷikʷəƛ̓əm), Squamish (Sḵwx̱wú7mesh Úxwumixw) and Musqueam (xʷməθkʷəy̓əm) Nations, on whose traditional and unceded territories Simon

Fraser University and our research program resides. This research was conducted across Canada and the United States, which include the traditional territories of many other Indigenous Peoples.

*Financial support:* KF received funding for this project from the Government of Canada Social Sciences and Humanities Research Council via a Joseph Armand Bombardier Canada Graduate Scholarship-Master's (CGS M). Public avalanche

safety research at Simon Fraser University Avalanche Research Program is further supported by Avalanche Canada and the Avalanche Canada Foundation.





## Appendix A

This appendix includes screen shots of all the bulletin scenarios with the solutions and explanations.


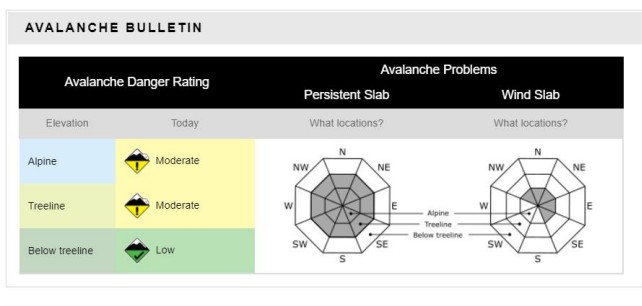

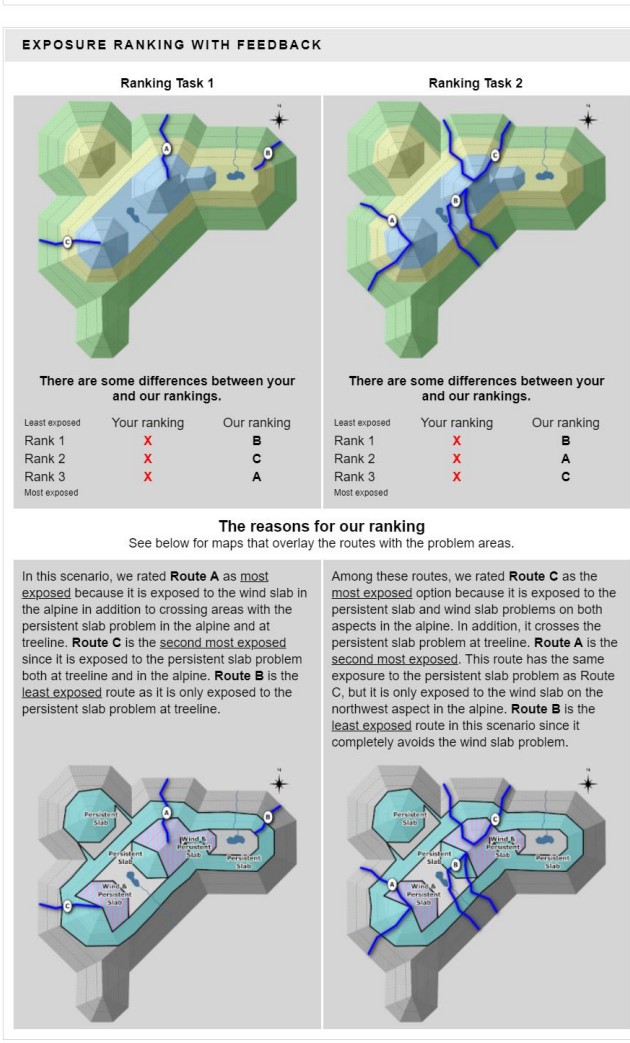

**Figure A1: Screen shot of Scenario 1 (ID 1) with avalanche bulletin information, route options, ranking solutions and explanations.**





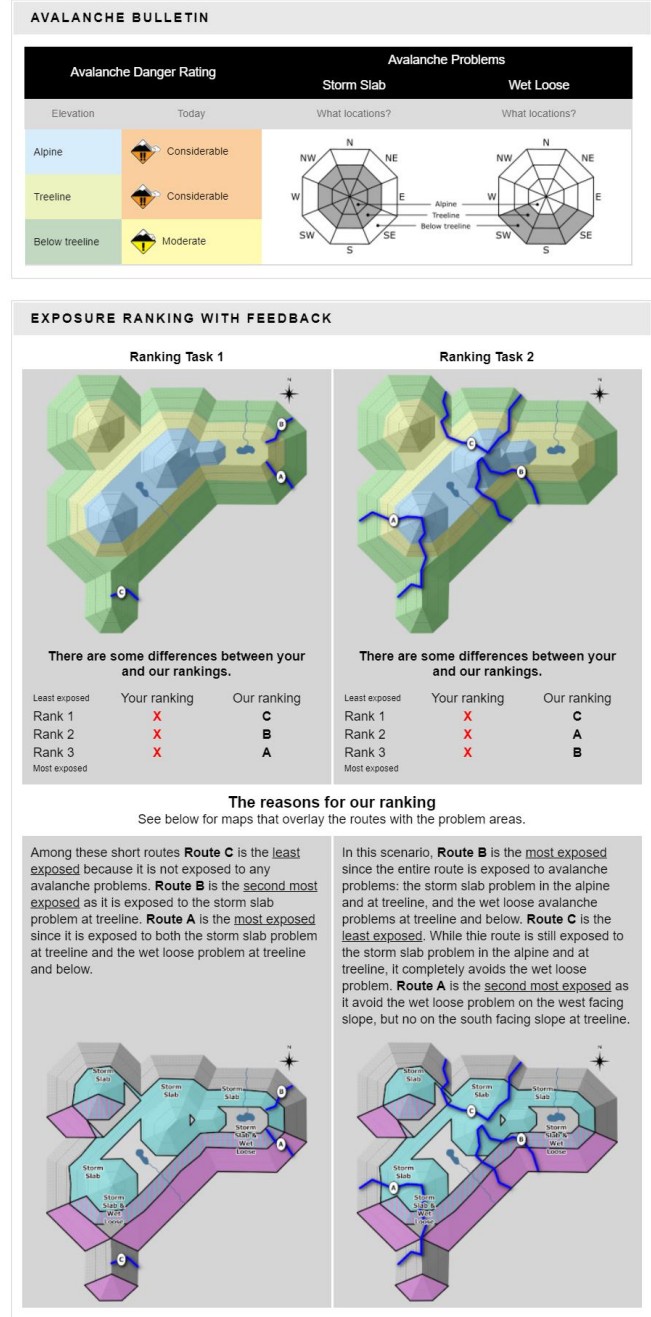

**Figure A2: Screen shot of Scenario 2 (ID 5) with avalanche bulletin information, route options, ranking solutions and explanations.**




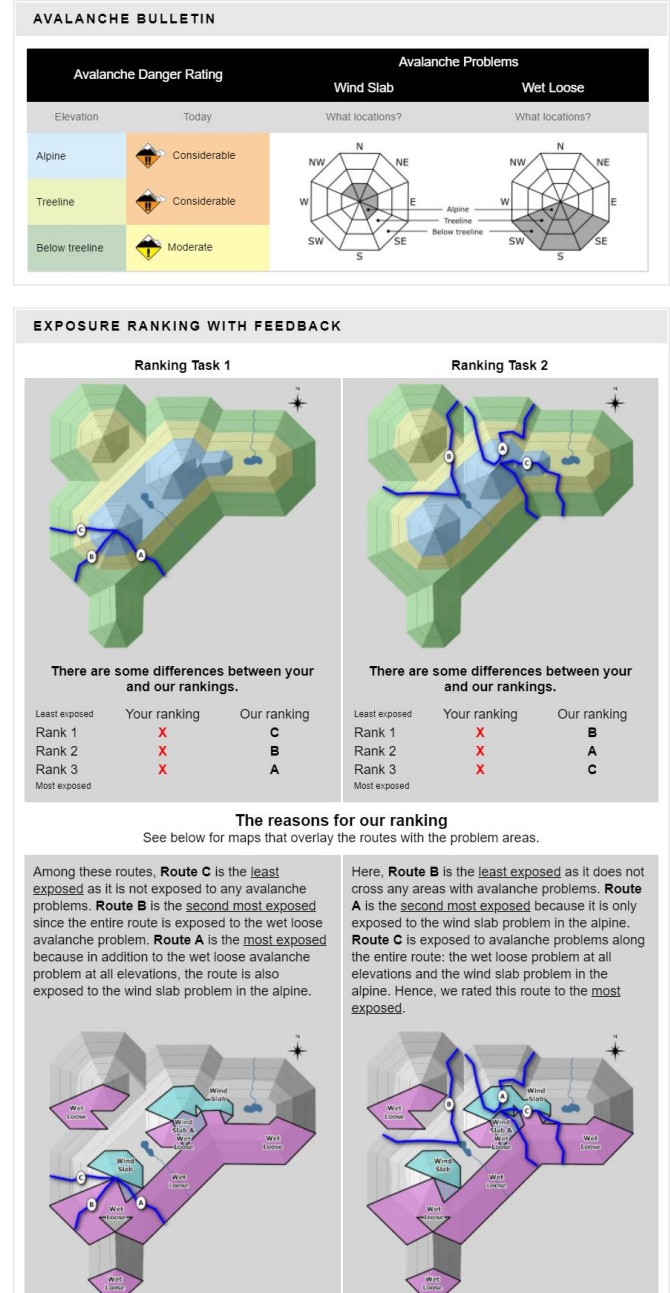

**Figure A3: Screen shot of Scenario 3 (ID 6) with avalanche bulletin information, route options, ranking solutions and explanations.**



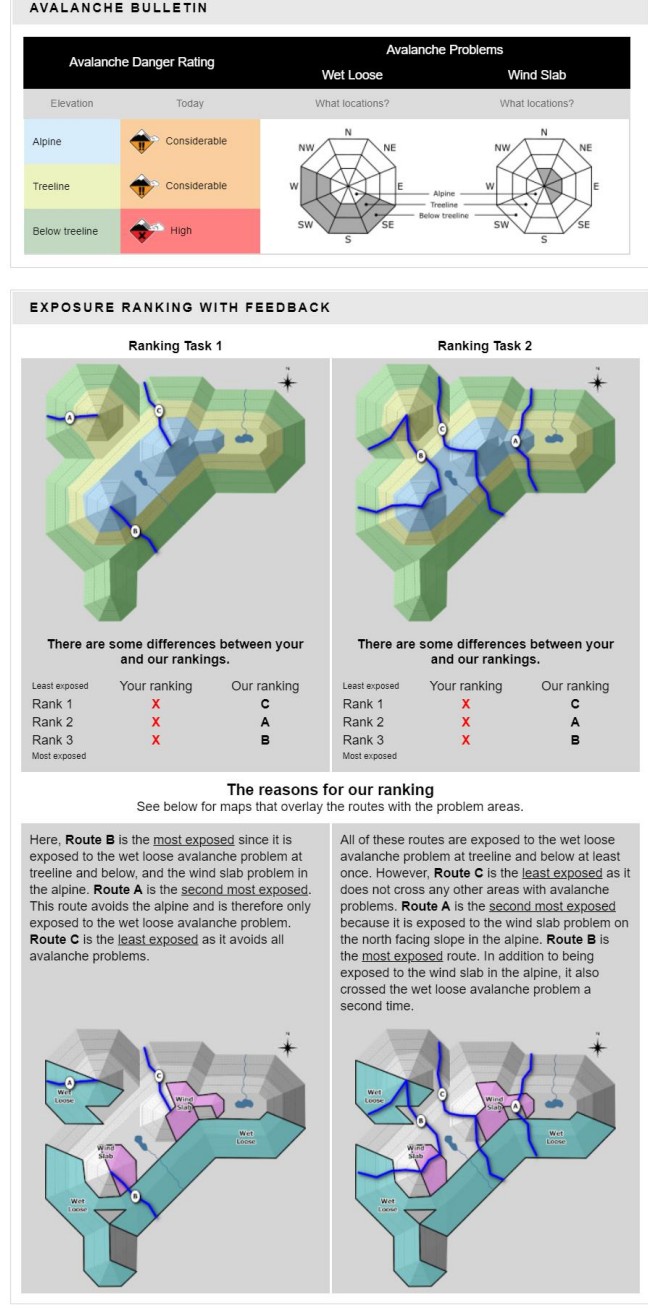

**Figure A4: Screen shot of Scenario 4 (ID 7) with avalanche bulletin information, route options, ranking solutions and explanations.**




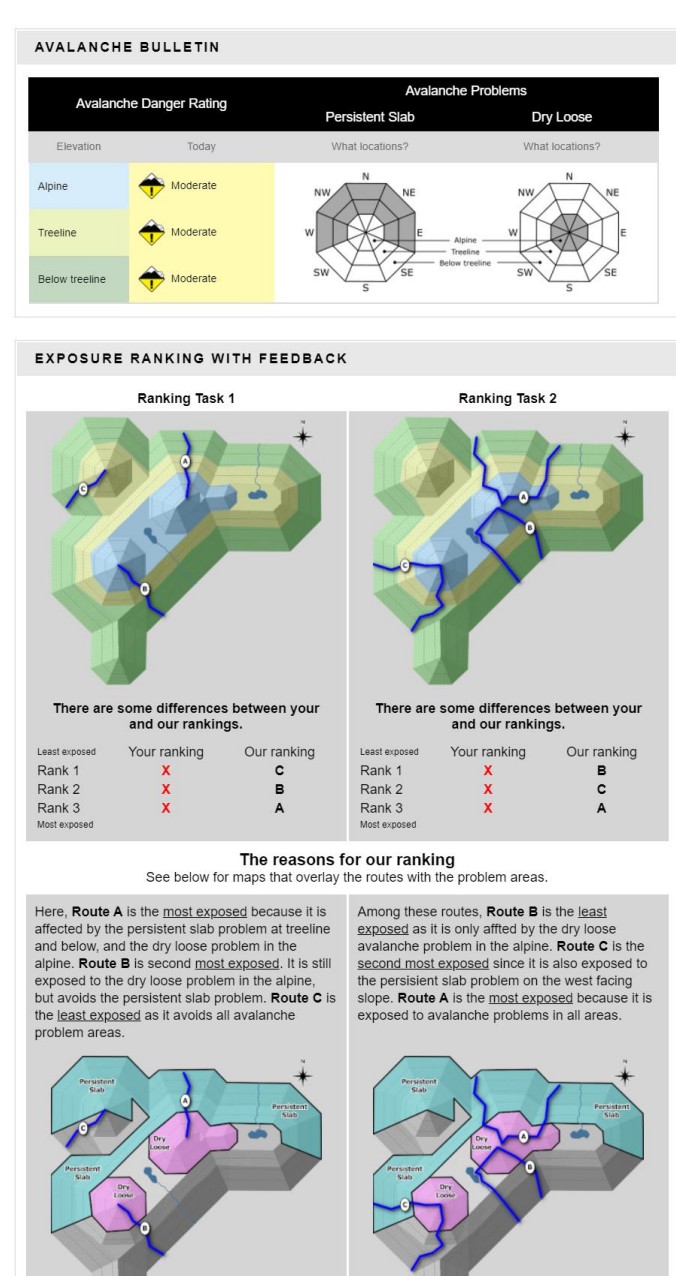

**Figure A5: Screen shot of Scenario 5 (ID 8) with avalanche bulletin information, route options, ranking solutions and explanations.**



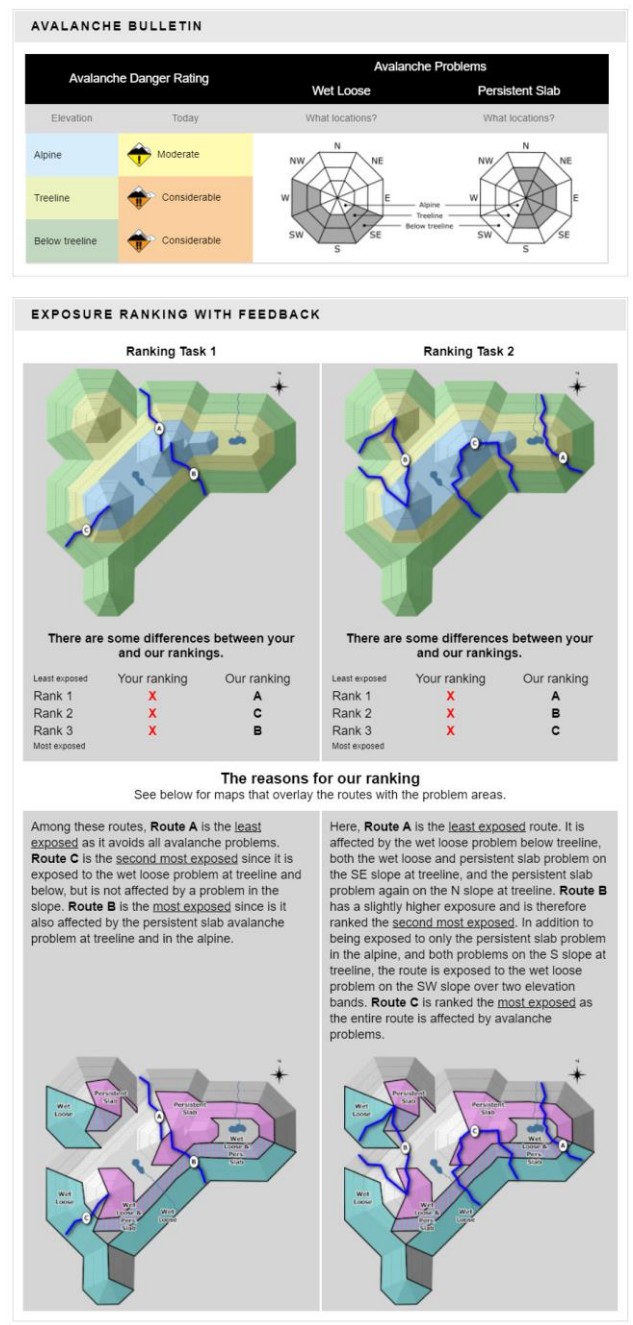

**Figure A6: Screen shot of Scenario 6 (ID 9) with avalanche bulletin information, route options, ranking solutions and explanations.**



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
