# Peer review of "Impact of information presentation on interpretability of spatial hazard information: Lessons from a study in avalanche safety"

_Natural Hazards and Earth System Sciences, 2021_

## Referee Comment (RC1)

**Review of «Impact of information presentation on interpretability of spatial hazard information: Lessons from a study in avalanche safety» by Fisher et al.**

The manuscript presents the results from a survey exploring the effectiveness of different graphic designs to communicate the location of avalanche problems. I consider the presented work of good quality requiring only minor revisions.

I particularly appreciate the well-designed structure of the survey, a structure which permitted to explore the influence of prior knowledge or previous experience a participant had with different graphical designs. The methods applied to analyze the responses are appropriate, the results are presented in an understandable way. Finally, I agree with the statement that the findings presented may help avalanche warning services to improve the way location information is communicated in avalanche forecasts.

I have a few minor recommendations / questions.

**1 General comments**

Numbers indicate line numbers.

**1.1 Survey Design**

As I said before, I consider the survey well-designed. However, there is one point, which I would like to address:

166-167: You state that participants were explicitly asked not to include the danger rating provided with the avalanche problem information when making their assessment. - I understand that the focus in this study was on applying location information relating to avalanche problems in the route-ranking task. However, I wonder, whether asking participants to ignore the danger rating in the assessment could convey a wrong message, namely that the location of the avalanche problems is more important than the severity of the hazard? After all, the severity of the avalanche hazard is summarized by the danger level, while the avalanche problems by themselves only answer the question «What is the problem?». The severity of the avalanche hazard, the danger level, also correlates much stronger with avalanche risk compared to the avalanche problem, and should therefore weighed more when planning routes[1].

For instance, in scenario 2, it seems that the storm-slab problem is much more severe (danger level *3 - Considerable* at the indicated elevation bands) - and thus the most relevant to consider, compared to the wet-loose problem (*2 - Moderate* below
* * *
[1]At least in Switzerland, we see a much stronger correlation between danger level and risk compared to the correlation between avalanche problems and risk (Techel et al., 2015; Winkler et al., 2021).

treeline). Another such example is scenario 4, where the wet-loose problem leads to a danger level *4 - High* below treeline, the wind-slab problem to a *3 - Considerable*. - I am not questioning the findings from your study, but I wonder why you chose this approach, and whether the danger rating, which was shown in the route-ranking exercise (Fig. 1) may actually have influenced some of the choices made. - Could you elaborate on this?

**1.2 Type of feedback given between route-ranking tasks**

The *type of feedback* provided between route-ranking tasks is mentioned several times (153, 186-187, ...). It is also explored as a main effect (Tab. 2, 394-396). - While I understand that the type of feedback will be analysed in a separate manuscript, maybe you could still provide a little more detail, otherwise the reader is left to speculate. Currently, the only explanation are the four classes mentioned on line 193.

**1.3 Recommendations regarding survey design**

Beside the actual findings regarding the design of location information in avalanche forecasts, I feel that the experience gained when designing this study may also help other avalanche warning services to design future user surveys, particularly when users from different forecast areas are expected to participate. - Maybe it would be worth adding some key recommendations in this regard in the Discussion section?

**2 Specific comments**

53-54: Rather than saying that European warning services use a *less-formalized terminology* to describe avalanche problems, I feel that the way of communicating the location and nature of avalanche problems *varies* in Europe. There are some warning services that use a highly formalized terminology to explain the location and nature of the avalanche problem. Such an example is the forecast in Norway, where the avalanche problem, the aspects and elevation, but also the expected avalanche size, the frequency of potential triggering spots and the trigger are specified (e.g. https://varsom.no/snoskredvarsling/varsel/Nord-Troms/ 2021-04-09). In contrast, there are many others where it is indeed much less formalized (e.g. in Switzerland or France).

53: note that in Europe the *avalanche problem types* are called *avalanche problems*

56: *structure* rather than *structures*?

72-82: maybe of interest in this context is the recent study by (Hutter et al., 2021), where the consistency in describing avalanche danger in public avalanche forecasts is analyzed for Swiss avalanche forecasts

112-128: Very useful overview on how aspect-elevation plots are currently used in forecast products.

112-137: Maybe of interest here (or in the Discussion section): In Europe, on the private www.skitourenguru.ch-platform, the information presented in the avalanche forecast (danger level, aspect and elevation) is directly applied to the map (considering aspect, elevation, terrain classification). Given a valid avalanche forecast, the user can obtain a rating for ski-touring routes. - Maybe it is worth mentioning that there are developments which go further than simply showing the location information using aspect-elevation graphs, as used in the public bulletins?

129-137: would it be possible to provide a figure with some examples to highlight the variety in graphical designs to support this section?

156: maybe add *(a)* after *the first research question*?

157: Would it be possible to provide references to these manuscripts already?

235: comma rather before *6,789* than afterwords?

286: the variable *n* is not explained

324: should it maybe read *consisted of 3,056 participants* rather than *consisted 3,056 participants*

378: the median and interquartile range of completion times are much lower than the mean and 95% confidence interval shown in Fig. 3 or the mean values mentioned later on in the text. - Can you please check whether this is an error. Maybe it would be helpful to report the mean value as well if it deviates a lot from the median.

403: should it read *without any training* rather than *with any training*

410: should it read *completed* rather than *complete*?

411: here is a reference to age classes, but these have not been introduced before - Consider presenting a small table in the Data Analysis section where all the variables and their classes are shown. This table could then also be used to highlight variables which were used in the models as main effects. I personally would have found this useful.

421-422: the p-values shown are $< 0.05$, which is considered significant (299); here it is described as *not significant* - please check this statement

428: I found this sentence hard to read. Is the first comma necessary?

428-429: In this sentence the name for the *Combined* is not in italics a is normally the case throughout the manuscript.

426: *decreased* rather than *decrease*?

445: *participants* rather than *participant*?

518: delete one of the two *that*

565: delete *for* prior to *during*?

**2.1 Figures**

The figures are clear and of good quality.

Please consider the following points as suggestions. However, I feel that addressing these could make it easier for the reader to link text statements to (sub)figures.

Fig. 1: maybe additionally show an example with complex routes beside this plot? This might make it easier understandable when you speak about simple and complex routes (185-186).

Fig. 2: could you maybe add *separate*, *aspect-elevation rose* and *combined* in the three subfigures?

Sect. 3.1: I personally would have liked a figure which would summarize the most important demographics of the participants, like the distribution of the classes for avalanche safety training, the type of activity, backcountry experience, and particularly bulletin user types. After all, and as you correctly point out, surveys tend to reach a very particular range of users.

Fig. 3: consider adding (a) to (d) to the four sub-figures. This would allow referencing the specific subfigure in the text.

Fig. 4: consider adding (a) to (c) to the three sub-figures. This would allow referencing the specific subfigure in the text.

90

**2.2 Tables**

As you write (307-308), the parameter estimates for the GLMM are hard to directly interpret due to the logit-link. - Maybe consider moving these tables to the appendix, and show plots with the main effects for all parameters instead (as you have already done in Fig.s 3 and 4 for some of the parameters)? As a reader, I found the Figures more helpful than the numbers

95   themselves.

I hope these comments help to improve the manuscript.

100   Frank Techel

**References**

Hutter, V., Techel, F., and Purves, R.: How is avalanche danger described in public avalanche forecasts?Analyzing textual descriptions of avalanche forecasts in Switzerland, Natural Hazards Earth System Sciences Discussion, 2021.

Techel, F., Zweifel, B., and Winkler, K.: Analysis of avalanche risk factors in backcountry terrain based on usage frequency and accident data in Switzerland, Nat. Hazards Earth Syst. Sci., 15, 1985–1997, https://doi.org/10.5194/nhess-15-1985-2015, 2015.

Winkler, K., Schmudlach, G., Degraeuwe, B., and Techel, F.: On the correlation between the forecast avalanche danger and avalanche risk taken by backcountry skiers in Switzerland, Cold Regions Science and Technology, 188, 103 299, https://doi.org/https://doi.org/10.1016/j.coldregions.2021.103299, https://www.sciencedirect.com/science/article/pii/S0165232X2100080X, 2021.

105

---

## Referee Comment (RC2)

*Review of «Impact of information presentation on interpretability of spatial hazard information: Lessons from a study in avalanche safety» by Fisher et al., [https://nhess.copernicus.org/preprints/nhess-2021-147](https://nhess.copernicus.org/preprints/nhess-2021-147)*

The manuscript is well written and includes a thorough analysis of a relatively large data set from a web-based survey – to test how effectively the aspect and elevation of avalanche problems may be communicated in avalanche forecasts.

The scope of manuscript is narrow, limited to testing how to graphically communicate two parameters related to avalanche problems. A broader scope could be beneficial, e.g. testing more options such as presenting the two properties on maps or including the results from testing interactive exercises.

However, the narrow scope facilitates a well-structured and easy to follow manuscript in all parts and I recommend it for publication in NHESS after minor revisions. It is well suited for publication in NHESS.

Specific comments, referring to line or chapter numbers in the manuscript:

Line 32: Suggest adding "may be" after risk

Chapter 3.1: I would appreciate a more comprehensive presentation of the participants, it would be nice to see the numbers for all types of activities, user types etc.

Chapter 3.2 and 4: A 0.7 probability of completing the task correctly suggest that many of the users would make the wrong decision about which route to choose. Even though it is positive that probabilities increase with training, it is worrying that it is not easier to make the right choice. I am missing a discussion about whether a probability of 0.7 indicate that the avalanche problem location is effectively communicated or rather calls for improved communication practices. It would be useful to elaborate on which other means could be available, e.g., could presenting the information about the avalanche problem location directly on the maps use in the route ranking test give better results. It would also be interesting to include a discussion on the fact that several warning services present the elevation limits in meters above sea level rather referring to the tree line – how may the results of this study be relevant and transferable to graphics which not only relate to the tree line?

Chapter 3: Chapter 3.2 and 3.3 analyse the results wrt. route ranking task completion correctness and completion time. Would it be possible and useful to analyse the data wrt. the primary activity of the participants in more detail? How are the results for snowmobilers vs. skiers? This is just an idea, if there is more of value to extract from the data set.

Line 403: I presume "out" should read "without"

Figure 4: The figures are slightly difficult to read, in particular the legend of the middle diagram. Using easily separable symbols for the different classes of points would improve the diagram.

*Rune Verpe Engeset, Oslo, 1 September 2021*

---

## Author Comment (AC1)

*Fisher et al.: Impact of information presentation on interpretability of spatial hazard information: Lessons from a study in avalanche safety*

This document includes our responses to the comments of both reviewers. In addition to addressing the concerns of the reviewers, we also carefully edited the entire manuscript.

**Response to Reviewer 1 (Frank Techel)**

September 18, 2021

***Reviewer comment:*** The manuscript presents the results from a survey exploring the effectiveness of different graphic designs to communicate the location of avalanche problems. I consider the presented work of good quality requiring only minor revisions. I particularly appreciate the well-designed structure of the survey, a structure which permitted to explore the influence of prior knowledge or previous experience a participant had with different graphical designs. The methods applied to analyze the responses are 5 appropriate, the results are presented in an understandable way. Finally, I agree with the statement that the findings presented may help avalanche warning services to improve the way location information is communicated in avalanche forecasts.

I have a few minor recommendations / questions.

***Author response:*** *Dear Frank: Thank you for your supportive and constructive comments.*

**General comments**

**1.1 Survey Design**

***Reviewer comment:*** As I said before, I consider the survey well-designed. However, there is one point, which I would like to address:

166-167: You state that participants were explicitly asked not to include the danger rating provided with the avalanche problem information when making their assessment. - I understand that the focus in this study was on applying location information relating to avalanche problems in the route-ranking task. However, I wonder, whether asking participants to ignore the danger rating in the assessment could convey a wrong message, namely that the location of the avalanche problems is more important than the severity of the hazard? After all, the severity of the avalanche hazard is summarized by the danger level, while the avalanche problems by themselves only answer the question «What is the problem?». The severity of the avalanche hazard, the danger level, also correlates much stronger with avalanche risk compared to the avalanche problem, and should therefore weighed more when planning routes1. For instance, in scenario 2, it seems that the storm-slab problem is much more severe (danger level 3 - Considerable at the indicated elevation bands) - and thus the most relevant to consider, compared to the wet-loose problem (2 - Moderate below treeline). Another such example is scenario 4, where the wet-loose problem leads to a danger level 4 - High below treeline, the wind-slab problem to a 3 - Considerable. - I am not questioning the findings from your study, but I wonder why you chose this approach, and whether the danger rating, 25 which was shown in the route-ranking exercise (Fig. 1) may actually have influenced some of the choices made. - Could you elaborate on this?

*Author response: We completely agree with Frank's comment that the danger rating is a critical part of the information presented in the avalanche bulletin that needs to be integrated into the assessment process when making decision about when and where to travel in the backcountry.*

*Asking participants assess and rank the routes' exposure to the presented avalanche problems instead of the severity of the avalanche conditions allowed us to eliminate the potential influence of participants' personal perceptions of the danger scale (e.g., is travelling in two sectors of moderate better or worse than one sector of considerable) and their individual risk propensities (e.g., I am comfortable with travelling in a sector with a considerable wind slab but not a moderate persistent slab). These personal aspects would have made it impossible to define objectively correct solutions for the task. Focusing exclusively on the exposure avoids these challenges and keeps the task as targeted to our research question as possible.*

*To explain the reasoning behind the design of our route ranking task in more detail, we have added the following explanations in Sections 2.1 (Survey design):*

*"Whereas examining only the exposure of the shown route does not fully represent the risk assessment process required for making informed trip planning decisions, our task design allowed us to eliminate any influences of participants' personal perception of the danger scale and their risk propensities in our experiment. In addition, it prevented us from having to quantify which avalanche problems were more or less hazardous under the same danger rating. All these aspects. would have made it impossible to define objectively correct solutions for the route ranking task and resulted in a much more challenging analysis."*

**1.2 Type of feedback given between route-ranking tasks**

**Reviewer comment:** The type of feedback provided between route-ranking tasks is mentioned several times (153, 186-187, …). It is also explored as a main effect (Tab. 2, 394-396). - While I understand that the type of feedback will be analysed in a separate manuscript, maybe you could still provide a little more detail, otherwise the reader is left to speculate. Currently, the only explanation are the four classes mentioned on line 193.

*Author response: The challenge is that explaining this additional aspect of our study in a meaningful way requires a considerable amount of detail. This is the reason why we are presenting the results in two different manuscripts. However, to give the reader of this manuscript a little bit more information, we have added the following section in Section 2.1 (Study design)*

*"Between the two avalanche bulletin scenarios, participants were presented with a range of different learning interventions to examine how an interactive exercise can affect participants' ability to apply the avalanche problem information to terrain. These learning interventions included a self-reflection exercise, showing participants the correct route ranking, and providing users with the correct route ranking and explaining it. However, this part of the experiment is not the focus of this manuscript. Interested readers are referred to Fisher, Haegeli and Mair (submitted) for a complete description of this part of our study."*

**1.3 Recommendations regarding survey design**

**Reviewer comment:** Beside the actual findings regarding the design of location information in avalanche forecasts, I feel that the experience gained when designing this study may also help other avalanche

warning services to design future user surveys, particularly when users from different forecast areas are expected to participate. - Maybe it would be worth adding some key recommendations in this regard in the Discussion section?

*Author response: Thank you for your complements on the design of our survey. While we agree with the importance of promoting good survey design in the avalanche safety research community, we are hesitant to include recommendations in this paper that go beyond the detailed description of the survey that is already included in the manuscript. There are several textbooks that do an excellent job describing best practices for survey design in general (e.g., Dillman et al., 2014; Vaske, 2008). In our opinion, there were not any particular survey design lessons that emerged from our study that could be described in the discussion section of this paper. However, a review paper on best practices in survey design for avalanche safety research could be an interesting idea for a future submission.*

*However, to give readers access to our complete survey design, we added the following sentence and reference at the end of Section 2.1 (Survey design)*

*"Interested readers are referred to Fisher (2021) for a complete description of our survey including screen shots."*

*Dillman, D. A., Smyth, J. D., & Christian, L. M. (2014). Internet, Phone, Mail, and Mixed-Mode Surveys: The Tailored Design Method. Hoboken, New Jersey: John Wiley & Sons.*

*Vaske, J. J. (2008). Survey Research and Analysis: Applications in Parks, Recreation and Human Dimensions. State College, PA: Venture Publishing Inc.*

**1.4 Avalanche problem terminology**

**Reviewer comment:** 53-54: Rather than saying that European warning services use a less-formalized terminology to describe avalanche problems, I feel that the way of communicating the location and nature of avalanche problems varies in Europe. There are some warning services that use a highly formalized terminology to explain the location and nature of the avalanche problem. Such an example is the forecast in Norway, where the avalanche problem, the aspects and elevation, but also the expected avalanche size, the frequency of potential triggering spots and the trigger are specified (e.g. https://varsom.no/ snoskredvarsling/varsel/Nord-Troms/ 2021-04-09). In contrast, there are many others where it is indeed much less formalized (e.g. in Switzerland or France).

53: note that in Europe the avalanche problem types are called avalanche problems

*Author response: We have adjusted the wording to reflect the diversity of approaches among European warning services. This revised section of the introduction now reads:*

*"European avalanche warning services utilize a smaller list of avalanche problem types (called avalanche problems in Europe) and take a range of approaches to explain the location and nature of the present problems—though overall the approaches tend to be similar to the conceptual model of avalanche hazard."*

**1.5 Typo**

**Reviewer comment:** 56: structure rather than structures?

*Author response: We believe that the plural form is actually correct in this sentence.*

**1.6 Recent research on avalanche bulletin quality**

**Reviewer comment:** 72-82: maybe of interest in this context is the recent study by (Hutter et al., 2021), where the consistency in describing avalanche danger in public avalanche forecasts is analyzed for Swiss avalanche forecasts

*Author response: Thanks for highlighting this new study to us. We have now included it in our description of the existing research. The revise section of the introduction now reads:*

*"Example studies of this research theme include Lazar et al. (2016) who presented public avalanche forecasters with a series of avalanche danger scenarios to see whether they interpret them the same, Techel et al. (2018) who examines the spatial consistency and bias of avalanche danger ratings in avalanche bulletins in the European Alps, Statham et al. (2018b), who studied the consistency of avalanche problem assessments among the warning services in the Canadian Rocky Mountains, Clark (2019) who studied the link between avalanche problem assessments and danger ratings in Canadian avalanche bulletins, and Hutter et al. (2021) who investigated the relationship between danger descriptions and avalanche danger rating in Swiss avalanche forecasts. All these studies highlighted considerable challenges and the need to improve the production of avalanche bulletins."*

**1.7 Overview of aspect-elevation plots**

**Reviewer comment:** 112-128: Very useful overview on how aspect-elevation plots are currently used in forecast products.

*Author response: Thank you very much for this complement.*

**1.8 Use of aspect and elevation information in decision aids**

**Reviewer comment:** 112-137: Maybe of interest here (or in the Discussion section): In Europe, on the private www.skitourenguru.ch-platform, the information presented in the avalanche forecast (danger level, aspect and elevation) is directly applied to the map (considering aspect, elevation, terrain classification). Given a valid avalanche forecast, the user can obtain a rating for ski-touring routes. - Maybe it is worth mentioning that there are developments which go further than simply showing the location information using aspect-elevation graphs, as used in the public bulletins?

*Author response: Thank you for reminding us of this type of decision aid that take advantage of the location information presented in avalanche bulletins. This comment also relates to Review Comment 2.3, which highlighted that additional interventions might be needed to help bulletin users to take full advantage of the location information provided in the bulletin. To address these comments, we added the following new paragraph in the discussion section.*

*"Despite the improved performance of participants with the Aspect-Elevation-Rose and the positive impact of avalanche awareness education, the fact that overall, only 74.6% of the route ranking tasks were completed correctly highlights that additional interventions might be necessary to help avalanche bulletin users make better use of the presented location information. Klassen (2012) highlighted that the next frontier of avalanche bulletins is to better assist users linking the hazard information to terrain, and the skitourenguru.ch web platform (Schmudlach & Köhler, 2016) is an example of a decision aid that automatically evaluates the severity of backcountry ski routes based on the current, location-specific avalanche hazard information presented in the bulletin. While these types of decision aid have great potential for helping backcountry recreationists avoid application mistakes and make better use of the*

*bulletin information, a detailed examination of how users interpret the severity ratings of the ski routes is critical for better understanding the advantages and disadvantages of the automated avalanche hazard information processing."*

**1.9 Figure with overview of graphical designs - Easy**

**Reviewer comment:** 129-137: would it be possible to provide a figure with some examples to highlight the variety in graphical designs to support this section?

*Author response: We added a new figure (see below) with a selection of graphical designs in the introduction section.*

[Figure]

**1.10 Editorial**

**Reviewer comment:** 156: maybe add (a) after the first research question?

*Author response: Instead of just adding an "(a)" at the end of this sentence to refer back to the research question, we simply repeated the entire wording to make it easier to the reader to understand the focus of this manuscript.*

**1.11 Reference for additional manuscripts**

**Reviewer comment:** 157: Would it be possible to provide references to these manuscripts already?

*Author response: We are currently in the process of finalizing these manuscripts for submission. We currently put placeholders for these references in the manuscript (Fisher, Haegeli & Mair, in prep a and b), and we will add the proper references as soon as they become available.*

**1.12 Grammar**

**Reviewer comment:** 235: comma rather before 6,789 than afterwords?

*Author response: Thank you for pointing out this typo.*

**1.13 Variable n**

**Reviewer comment:** 286: the variable n is not explained

**Author response:** *n represents that number of observations. This explanation has been added to the text.*

**1.14 Typo**

**Reviewer comment:** 324: should it maybe read consisted of 3,056 participants rather than consisted 3,056 participants

**Author response:** *Thank you for pointing out this typo.*

**1.15 Median and IQR of completion times**

**Reviewer comment:** 378: the median and interquartile range of completion times are much lower than the mean and 95% confidence interval shown in Fig. 3 or the mean values mentioned later on in the text. - Can you please check whether this is an error. Maybe it would be helpful to report the mean value as well if it deviates a lot from the median.

**Author response:** *The median completion time presented at the beginning of Section 3.3 is fundamentally different from the completion time figures presented in the effects plots in Figure 3 (now Figure 5) and the associated text. The median completion time is the straight up median calculated from the raw survey data. Since we are dealing with a distribution that is bound by zero on the left and has a long tail on the right, the median and interquartile range are the more appropriate summary statistics than the mean and standard deviation.*

*The effect plots shown in Fig.3 show estimated marginal means of the completion times (and 95% confidence interval) based on the model result for a specific combination of predictor values. For example, Fig. 5c shows the completion time as function of the information format with all other predictors fixed at their base level. Hence, these plots are used to illustrate the effect of individual predictors and do not represent the dataset as a whole. What the effects plots show was already explained at the very end of Section 3.2 (Data analysis). However, we slightly expanded the text to make the explanation clearer. The text no reads:*

*"The results of these analyses are presented in so-called effects plots, which display the differences between levels of a predictor variable of interest while holding all other predictor variables constant at their base levels. Hence, it is more important to look at the differences between the attribute levels of the predictor variable of interest than the absolute values since these charts simply illustrate the magnitude of the effect of the predictor variable and do not provide an overview of the overall nature of the dataset."*

**1.16 Typo**

**Reviewer comment:** 403: should it read without any training rather than with any training

**Author response:** *Thank you for pointing out this typo.*

**1.17 Typo**

**Reviewer comment:** 410: should it read completed rather than complete?

**Author response:** *Thank you for pointing out this typo.*

**1.18 Table of demographic variables**

**Reviewer comment:** 411: here is a reference to age classes, but these have not been introduced before - Consider presenting a small table in the Data Analysis section where all the variables and their classes are shown. This table could then also be used to highlight variables which were used in the models as main effects. I personally would have found this useful.

*Author response: The better illustrate the nature of our survey sample, we have added a new figure (see below) that shows the distributions of the different demographic variables included in our analysis. This also addresses Reviewer comments 1.26 and 2.2.*

[Figure]

**1.19 Significant statement**

**Reviewer comment:** 421-422: the p-values shown are < 0.05, which is considered significant (299); here it is described as not significant – please check this statement

*Author response: Thank you for catching this typo!*

**1.18 Readability**

**Reviewer comment:** 428: I found this sentence hard to read. Is the first comma necessary?

*Author response: Thank you for pointing out this typo.*

**1.19 Formatting**

**Reviewer comment:** 428-429: In this sentence the name for the Combined is not in italics a is normally the case throughout the manuscript.

*Author response:* *Thanks for pointing out this formatting error. We fixed the formatting of 'Combined'.*

**1.20 Typo**
*Reviewer comment:* 426: decreased rather than decrease?

*Author response:* *We found this typo on line 436, where we fixed it.*

**1.21 Typo**
*Reviewer comment:* 445: participants rather than participant?

*Author response:* *Thank you for pointing out this typo.*

**1.22 Typo**
*Reviewer comment:* 518: delete one of the two that

*Author response:* *Thank you for pointing out this typo.*

**1.23 Typo**
*Reviewer comment:* 565: delete for prior to during?

*Author response:* *Thank you for pointing out this typo.*

**Figures**
*Reviewer comment:* The figures are clear and of good quality. Please consider the following points as suggestions. However, I feel that addressing these could make it easier for the reader to link text statements to (sub)figures.

**1.24 Figure 1**
*Reviewer comment:* Fig. 1: maybe additionally show an example with complex routes beside this plot? This might make it easier understandable when you speak about simple and complex routes (185-186).

*Author response:* *We have updated the figure to include both simple and complex routes (see below)*

[Figure]

a) Simple Routes    b) Complex Routes

**1.25 Figure 2**

**Reviewer comment:** Fig. 2: could you maybe add separate, aspect-elevation rose and combined in the three subfigures?

**Author response:** *We have updated the figure (see below).*

a) Separate Graphics    b) Aspect-Elevation Rose    c) Combined Graphic

[Figure]

**1.26 Additional figure on demographics**

**Reviewer comment:** Sect. 3.1: I personally would have liked a figure which would summarize the most important demographics of the participants, like the distribution of the classes for avalanche safety training, the type of activity, backcountry experience, and particularly bulletin user types. After all, and as you correctly point out, surveys tend to reach a very particular range of users.

*Author response: We added the requested figure. See our response to reviewer comment 1.18 for more details.*

**1.27 Figure 3**

**Reviewer comment:** Fig. 3: consider adding (a) to (d) to the four sub-figures. This would allow referencing the specific subfigure in the text.

*Author response: We have updated the figure as requested (see below) and referenced the sub-figures in the text.*

[Figure]

**1.28 Figure 4**

**Reviewer comment:** Fig. 4: consider adding (a) to (c) to the three sub-figures. This would allow referencing the specific subfigure in the text.

*Author response: We have updated the figure as requested (see below) and referenced the sub-figures in the text.*

[Figure]

**Tables**

**1.29 Additional figures to visualize main effects - Medium**

**Reviewer comment:** As you write (307-308), the parameter estimates for the GLMM are hard to directly interpret due to the logit-link. – Maybe consider moving these tables to the appendix, and show plots with the main effects for all parameters instead (as you have already done in Figs 3 and 4 for some of the parameters)? As a reader, I found the Figures more helpful than the numbers themselves.

*Author response: Fundamentally, we very much agree with Frank's opinion that plot can provide a more easily digestible perspective on the model characteristics than the raw regression parameters. However, there are several complications that make us hesitant to follow the reviewers' recommendation to replace the existing tables with effects plots. First, we believe that the tables should be included as they present the results of the regression analysis in the most complete way. Second, effects plots only provide a very specific perspective on a model by highlighting the effect of a single predictor when holding all other predictors at a specific level (typically the average or the base level). Hence, the values of the dependent variable shown on the y-axis in an effects plot are only representative for the specified combination of predictor values and not for the dataset as a whole (see also our response to comment 1.15). Furthermore, the confidence intervals shown in the effects plots do not show whether parameter estimates are significantly different from each other. This means that while effects plots are visually intuitive, they need considerable explanations to avoid misunderstandings. Hence, we believe they are best used to illustrate specific research questions. The general trends whether a particular predictor has a positive or negative relationship with the dependent variable can be extracted from the parameter estimates in the table directly. Third, given that our manuscript presents three regression models with many parameters, we believe that the number of plots would become overwhelming and drown the key messages we are conveying with the plots currently included. We believe that the current combination of the tables and the select plots allows us to present the complete results and highlight the patterns relevant for answering the research questions in the most effective way. Hence, we would prefer to stay with the current selection of tables and effects plots in the main body of the manuscript.*

**Response to Reviewer 2 (Rune Engeset)**

September 18, 2021

***Reviewer comment:*** The manuscript is well written and includes a thorough analysis of a relatively large data set from a web-based survey – to test how effectively the aspect and elevation of avalanche problems may be communicated in avalanche forecasts.

The scope of manuscript is narrow, limited to testing how to graphically communicate two parameters related to avalanche problems. A broader scope could be beneficial, e.g. testing more options such as presenting the two properties on maps or including the results from testing interactive exercises.

However, the narrow scope facilitates a well-structured and easy to follow manuscript in all parts and I recommend it for publication in NHESS after minor revisions. It is well suited for publication in NHESS.

***Author response:*** *Dear Rune: Thank you for your supportive and constructive comments.*

**2.1 Wording modification**

***Reviewer comment:*** Line 32: Suggest adding "may be" after risk

***Author response:*** *Thanks for highlighting this as it certainly not a given that recreationists manage avalanche risk in the described way. This sentence now reads:*

*"When travelling in the backcountry avalanche risk is ideally managed by carefully assessing the nature and severity of the hazard using weather, snowpack and avalanche observations (e.g., McClung, 2002)."*

**2.2 More details on survey participants**

***Reviewer comment:*** Chapter 3.1: I would appreciate a more comprehensive presentation of the participants, it would be nice to see the numbers for all types of activities, user types etc.

***Author response:*** *This comment is in line with comments 1.18 and 1.25 of Reviewer #1. In response, we added a new figure that illustrates the nature of our survey sample in more detail. See our response to comment 1.18 for more details.*

**2.3 Discussion of overall performance in route ranking**

***Reviewer comment:*** Chapter 3.2 and 4: A 0.7 probability of completing the task correctly suggest that many of the users would make the wrong decision about which route to choose. Even though it is positive that probabilities increase with training, it is worrying that it is not easier to make the right choice. I am missing a discussion about whether a probability of 0.7 indicate that the avalanche problem location is effectively communicated or rather calls for improved communication practices. It would be useful to elaborate on which other means could be available, e.g., could presenting the information about the avalanche problem location directly on the maps use in the route ranking test give better results.

***Author response:*** *This is an excellent point and offers a chance to comment on future research and development opportunities. This comment also relates to Review Comment 1.8, which highlighted that the skitourenguru.ch web platform is a decision aid that helps recreationists apply terrain information*

*and evaluate route options. To address these two comments, we added the following new paragraph in the discussion section.*

*"Despite the improved performance of participants with the Aspect-Elevation-Rose and the positive impact of avalanche awareness education, the fact that overall, only 74.6% of the route ranking tasks were completed correctly highlights that additional interventions might be necessary to help avalanche bulletin users make better use of the presented location information. Klassen (2012) highlighted that the next frontier of avalanche bulletins is to better assist users linking the hazard information to terrain, and the skitourenguru.ch web platform (Schmudlach & Köhler, 2016) is an example of a decision aid that automatically evaluates the severity of backcountry ski routes based on the current, location-specific avalanche hazard information presented in the bulletin. While these types of decision aid have great potential for helping backcountry recreationists avoid application mistakes and make better use of the bulletin information, a detailed examination of how users interpret the severity ratings of the ski routes is critical for better understanding the advantages and disadvantages of the automated avalanche hazard information processing."*

*Our addition to the limitation section (primarily relates to Reviewer Comment 2.4) also highlights that the effectiveness of automated route severity ratings should be explore in future research.*

*"Since this study focused primarily on a North American audience and our survey design did not include presentation formats with variable elevation values commonly used in European avalanche bulletins, the recommendations of our study should be applied with caution. Future research in this area should test a wider range of presentation formats including the European location graphics, the direct presentation of hazard locations on maps, and automated route severity ratings."*

**2.4 Extrapolation of results to European avalanche bulletins**

**Reviewer comment:** It would also be interesting to include a discussion on the fact that several warning services present the elevation limits in meters above sea level rather referring to the tree line – _how may the results of this study be relevant and transferable to graphics which not only relate to the tree line?

*Author response: This is an interesting line of thought. We were primarily focusing on a North American context, but it would be interesting to consider this in future work. To highlight this limitation of our study and point out future research opportunities, we added the following text in the first paragraph of Section 4.2 (Implications for avalanche warning services) and Section 4.3 (Limitations)*

*Section 4.2 (Implications for avalanche warning services):*
*"It is important to remember, however, that the location information presented in our survey used predefine elevation bands, and it is unclear whether the Aspect-Elevation Rose graphic is also the prefer presentation format with variable elevation values commonly used by European avalanche warning services. Still, the cognitive load perspective indicates that having Separate graphics with variable elevation values would likely results in higher extrinsic load than Separate graphics with static elevation values, and we therefore expect that presentation format to be even more challenging and error prone."*

*Section 4.3 (Limitations)*
*"Since this study focused primarily on a North American audience and our survey design did not include presentation formats with variable elevation values commonly used in European avalanche bulletins, the*

*recommendations of our study should be applied with caution. Future research in this area should test a wider range of presentation formats including the European location graphics, the direct presentation of hazard locations on maps, and automated route severity ratings."*

**2.5. Effect of primary activity**

**Reviewer comment:** Chapter 3: Chapter 3.2 and 3.3 analyse the results wrt. route ranking task completion correctness and completion time. Would it be possible and useful to analyse the data wrt. the primary activity of the participants in more detail? How are the results for snowmobilers vs. skiers? This is just an idea, if there is more of value to extract from the data set.

*Author response: While primary activity emerged as a significant predictor variable in our model for completion time (Table 1), we do not believe that discussing the effect in more detail brings much value to the manuscript. Our survey sample is highly dominated by backcountry skiers, and the design of the routes included in the route ranking task was also optimized for this activity. Hence, we do not believe that these differences, even though statistically significant, are informative. However, better highlight this limitation of our study, we expanded the following text to Section 4.2 (Implications for avalanche warning services) with additional information about the limitations of the activity specific information:*

*"Additionally, this study found that participants with different primary backcountry activities performed differently on the task exercises even after controlling for avalanche awareness training. However, there was no interaction effect between the type of avalanche problem graphic used and participants' primary backcountry activity, indicating that the graphic use was not a factor in this variation of performance. Avalanche warning services can use this as evidence that changing avalanche problem graphics will not disadvantage backcountry recreationists of any sport. However, even though the survey was open to all winter backcountry recreationists, most participants were backcountry skiers, and the routes shown in the ranking tasks were optimized to be realistic for backcountry skiing. This means that the route ranking exercise may have not fully resonated with other activity groups, such as snowmobilers, snowshoers, or ice climbers. Hence the results presented in this study should only be extrapolated to these user groups with caution. To better understand the skills and perspectives of all types of avalanche bulletin users, future studies should seek to create hypothetical terrain scenarios tailored to a wider range of backcountry activities. Additional research is also needed to determine if the effects observed during this desktop exercise can be translated"*

**2.6 Typo**

**Reviewer comment:** Line 406: I presume "out" should read "without"

*Author response: Thank you for pointing out this typo.*

**2.7 Figure 4**

**Reviewer comment:** Figure 4: The figures are slightly difficult to read, in particular the legend of the middle diagram. Using easily separable symbols for the different classes of points would improve the diagram

*Author response: Thank you for the feedback. We have updated the figures as requested. We are now using separate symbols for each class, and we have improved the placement of the labels in the figure.*